# *Mycobacterium tuberculosis* SatS is a chaperone for the SecA2 protein export pathway

Brittany K Miller[1], Ryan Hughes[2], Lauren S Ligon[1], Nathan W Rigel[1†], Seidu Malik[1], Brandon R Anjuwon-Foster[1‡], James C Sacchettini[2], Miriam Braunstein[1]*

[1]Department of Microbiology and Immunology, University of North Carolina at Chapel Hill, North Carolina, United States; [2]Department of Biochemistry and Biophysics, Texas A&M University, College Station, United States

*For correspondence:
braunste@med.unc.edu

Present address: †Department of Biology, Hofstra University, New York, United States; ‡Laboratory of Molecular Biology, National Cancer Institute, Bethesda, United States

Competing interests: The authors declare that no competing interests exist.

**Abstract** The SecA2 protein export system is critical for the virulence of *Mycobacterium tuberculosis*. However, the mechanism of this export pathway remains unclear. Through a screen for suppressors of a *secA2* mutant, we identified a new player in the mycobacterial SecA2 pathway that we named SatS for SecA2 (two) Suppressor. In *M. tuberculosis*, SatS is required for the export of a subset of SecA2 substrates and for growth in macrophages. We further identify a role for SatS as a protein export chaperone. SatS exhibits multiple properties of a chaperone, including the ability to bind to and protect substrates from aggregation. Our structural studies of SatS reveal a distinct combination of a new fold and hydrophobic grooves resembling preprotein-binding sites of the SecB chaperone. These results are significant in better defining a molecular pathway for *M. tuberculosis* pathogenesis and in expanding our appreciation of the diversity among chaperones and protein export systems.
DOI: https://doi.org/10.7554/eLife.40063.001

## Introduction

With 1.7 million deaths from tuberculosis in 2016, *Mycobacterium tuberculosis* continues to have a significant impact on world health (*World Health Organization, 2017*). For *M. tuberculosis* to cause disease, the bacillus must export effector proteins to the host-pathogen interface. These effectors enable *M. tuberculosis* to grow in macrophages and avoid clearance by the host immune response (*Awuh and Flo, 2017*). At least some of these effectors are exported by *M. tuberculosis* via the specialized SecA2 export pathway (*Sullivan et al., 2012*).

The mechanism of SecA2 export remains poorly understood. SecA2 is a paralog of the SecA ATPase of the general Sec protein export pathway. The general Sec pathway transports preproteins with N-terminal signal sequences across the inner membrane through a channel comprised of SecY, SecE and SecG proteins (*Brundage et al., 1990*). Preproteins must be in an unfolded state to travel through the SecYEG channel and, in Gram-negative bacteria, the SecB chaperone binds a subset of preproteins to maintain them in an unfolded translocation competent state. Following export across the membrane, the signal sequence is cleaved and the mature protein is released (*Tsirigotaki et al., 2017*). While all bacteria possess an essential Sec pathway that carries out the majority of protein export, only mycobacteria and a subset of Gram-positive bacteria possess specialized Sec export systems that are defined by a second SecA (*Bensing et al., 2014*; *Miller et al., 2017*). In these organisms, SecA1 is the name given to the canonical SecA and the specialized SecA is named SecA2. For the mycobacterial SecA2 system, the housekeeping SecYEG channel, and possibly SecA1, as well, are also involved (*Ligon et al., 2013*; *Prabudiansyah et al., 2015*). However, SecA1

and SecA2 are functionally distinct, as shown by their inability to compensate for the loss of one another (*Braunstein et al., 2001*; *Rigel et al., 2009*), and it remains unclear how SecA2 functions to export its relatively small and specific subset of proteins.

Here, we carried out a suppressor screen using a dominant negative *secA2 K129R* mutant of *Mycobacterium smegmatis*, a fast-growing model mycobacteria, as a means to identify new components of the mycobacterial SecA2 pathway. The K129R substitution is in the ATP binding site of SecA2, and past studies lead to a model where SecA2 K129R is defective for SecA2-dependent export but still able to interact with its normal binding partners that include SecYEG (*Rigel et al., 2009*; *Ligon et al., 2013*). As a result, SecA2 K129R disrupts SecYEG channels at the membrane, which hinders both general Sec and specialized SecA2 export as evidenced by more severe phenotypes of *secA2 K129R* than a *ΔsecA2* null mutation (*Ligon et al., 2013*). A large collection of *secA2 K129R* suppressor mutations mapped to *msmeg_1684*, a gene of unknown function that we renamed *satS* for SecA2 (two) Suppressor. SatS is also present in *M. tuberculosis* and, remarkably, the *M. tuberculosis satS* gene is in an operon with the gene encoding SapM, which is a secreted phosphatase exported by the SecA2 pathway (*Zulauf et al., 2018*).

Here, we demonstrated that SatS, which we revealed is required for *M. tuberculosis* growth in macrophages, functions in the export of SapM and an additional subset of the proteins exported by the SecA2 pathway. We further identified properties of SatS that indicate a function as a protein export chaperone that protects its substrates from inappropriate interactions in the cytoplasm and additionally assists in their export. Finally, we determined the structure of the C-domain of SatS (SatS$_C$), which reveals a new fold lacking similarities to any solved chaperone structures, yet contains surface hydrophobic grooves resembling those of the SecB chaperone. The identification of SatS expands our understanding of SecA2 export in mycobacteria and provides another example of the diversity of molecular chaperones across biological systems.

## Results

### *satS* suppressors of *secA2 K129R*

A *secA2 K129R* mutant of *M. smegmatis* exhibits more exacerbated phenotypes (*i.e.* azide sensitivity and poor growth on Mueller-Hinton agar) than a *ΔsecA2* deletion mutant (*Ligon et al., 2013*) (*Figure 1A*). Starting with cultures of *ΔsecA2* expressing the *secA2 K129R* allele on an integrating plasmid (this strain is referred to as *secA2 K129R* from hereon), we collected spontaneous suppressor mutants that restored growth on Mueller-Hinton agar. Whole-genome sequencing of six extragenic suppressors revealed mutations in the same gene *msmeg_1684* (*Figure 1B*). Three additional suppressors with mutations in *msmeg_1684* were identified by directly sequencing the *msmeg_1684* gene and upstream sequence in our pool of suppressors (*Figure 1B*). Msmeg_1684 is a highly acidic protein (pI 3.83) of unknown function with conserved homologs in all mycobacterial species, as well as other actinomycetes (*Marmiesse et al., 2004*). However, no homologous proteins exist outside of actinomycetes and Msmeg_1684 does not have any conserved domains to predict function. Henceforth, we refer to *msmeg_1684* as *satS* (secA2 (two) suppressor).

Seven of the nine suppressor mutations in *satS* were expected to be loss-of-function mutations (*i.e* frameshifts or truncations). To validate that loss of *satS* suppresses *secA2 K129R* phenotypes, we deleted *satS* in the *secA2 K129R* mutant background. For future experiments, we also constructed a *ΔsatS* mutant in a *secA2$^+$* background. The *ΔsatS* mutant had no in vitro growth defect compared to wild-type *M. smegmatis* mc$^2$155 (*Figure 1—figure supplement 1A*). Deletion of *satS* suppressed the exacerbated phenotypes of *secA2 K129R*, and the suppression phenotype of *ΔsatS* could be complemented by adding back a copy of *satS* from *M. smegmatis* (*Figure 1C*). Complementation was also successful with the *M. tuberculosis satS* homolog *rv3311* indicating that SatS function is conserved in *M. smegmatis* and *M. tuberculosis* (*Figure 1C*).

Unlike wild-type SecA2, which is predominantly in the cytoplasm, the majority of SecA2 K129R is localized to the membrane-containing cell envelope fraction, consistent with SecA2 K129R being trapped in a non-functional complex with SecYEG (*Rigel et al., 2009*). We assessed the ability of *ΔsatS* to suppress the mislocalization of SecA2 K129R using immunoblot analysis of envelope (cell wall and membrane) and soluble (cytoplasm) fractions with SecA2 antibodies. Total SecA2 K129R levels were unchanged in the *ΔsatS* background (*Figure 1D*). However, the absence of SatS

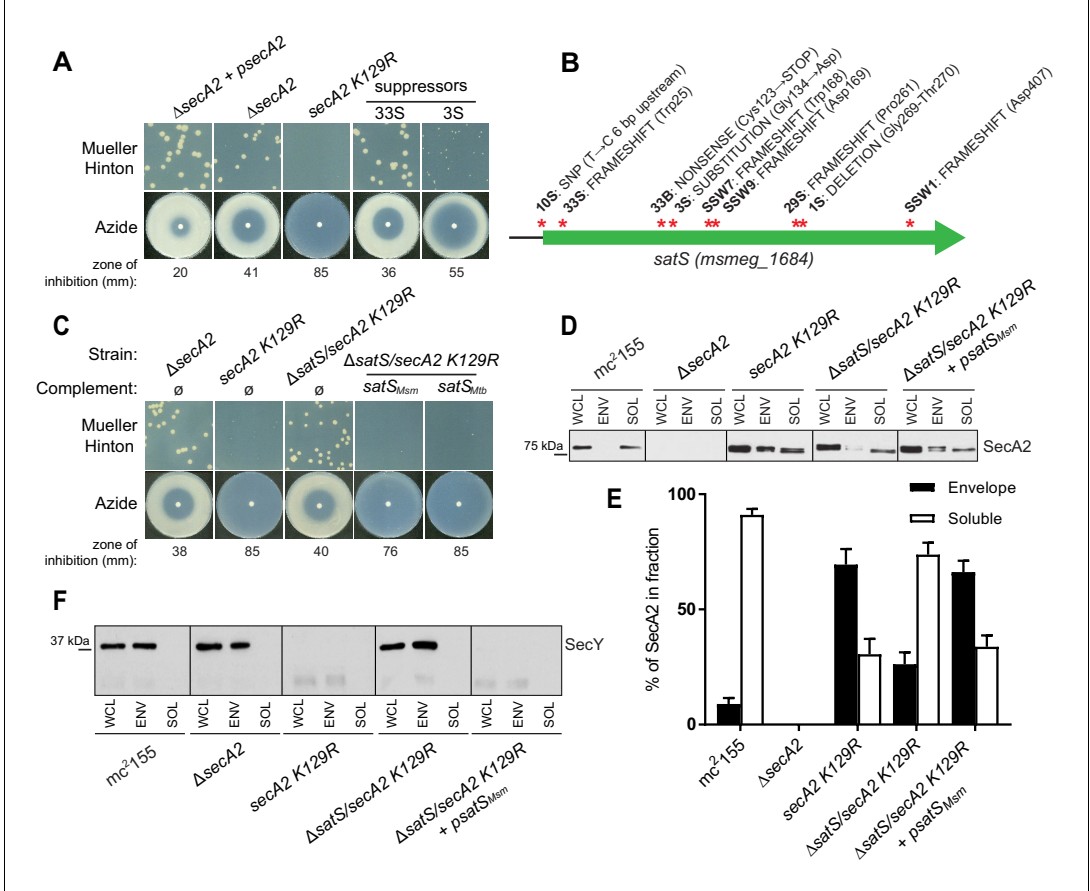

**Figure 1.** Δ*satS* mutant suppresses SecA2 K129R phenotypes. (**A**) Mueller-Hinton growth phenotypes and azide sensitivity of *M. smegmatis* Δ*secA2* mutant expressing wild-type *secA2* (Δ*secA2* +*psecA2*), an empty vector, or *secA2 K129R* (*secA2 K129R*), and two representative suppressors. (**B**) Nine suppressor mutations affected *satS* (*msmeg_1684*). Red stars indicate approximate locations of mutations. For panels C, D, E and F, wild-type *M. smegmatis* mc²155, Δ*secA2*, *secA2 K129R*, Δ*satS/secA2 K129R*, and Δ*satS/secA2 K129R* complemented with *satS_Msm* or *satS_Mtb* were used. (**C**) Mueller-Hinton growth phenotypes and azide sensitivity of the strains described above. (**D**) Whole cell lysates (WCL), subcellular envelope (ENV) and soluble (SOL) fractions were separated by SDS-PAGE, and SecA2 protein was detected by Immunoblot. (**E**) Densitometry was used to quantify SecA2 levels in the soluble and envelope fractions (ImageJ). Percent localization to a given fraction for SecA2 is reported as the percentage of the total (soluble +envelope). Error bars indicate the standard error of the mean of three independent experiments. (**E**) Subcellular fractions were separated by SDS-PAGE and SecY protein was detected by Immunoblot. All results shown are representative of at least three independent experiments.

DOI: https://doi.org/10.7554/eLife.40063.002

The following figure supplement is available for figure 1:

**Figure supplement 1.** Characterization of the *M. smegmatis* Δ*satS* mutant.

DOI: https://doi.org/10.7554/eLife.40063.003

suppressed the aberrant localization of SecA2 K129R (*i.e.* in the Δ*satS/secA2 K129R* strain) such that SecA2 K129R was now primarily localized to the cytoplasm, similar to wild-type SecA2 (*Figure 1D and E*). We immunoblotted for the cell wall MspA porin and the cytoplasmic GroEL protein as fractionation controls (*Figure 1—figure supplement 1B*). SecA2 K129R is also associated with reduced levels of SecY, which is a presumed mechanism to eliminate jammed SecA2 K129R-SecYEG channels (*Ligon et al., 2013*). When we immunoblotted fractions from the Δ*satS/secA2 K129R* strain with SecY antibodies, we observed that the absence of SatS suppressed the SecA2 K129R effect on SecY. Both the rescued localization of SecA2 K129R and SecY levels observed in the Δ*satS* mutant could be complemented by introduction of *satS_Msm* (*Figure 1D, E and F*). These results indicate that SatS is required for SecA2 K129R retention at the membrane in non-productive complexes with SecYEG. By extension, these results suggest a role for SatS in the SecA2 export pathway.

## SatS is required for export of the SecA2-dependent SapM phosphatase

In *M. tuberculosis*, the gene encoding SatS is immediately downstream of the gene encoding SapM. Reverse transcriptase (RT) PCR performed on RNA from wild-type *M. tuberculosis* strain H37Rv was used to demonstrate that *sapM* and *satS* are in an operon (*Figure 2—figure supplement 1*). This genomic arrangement is striking as SapM, a secreted phosphatase of *M. tuberculosis* is exported by the SecA2 pathway (*Zulauf et al., 2018*). While SapM does not have an ortholog in *M. smegmatis*, we identified 26 mycobacterial species in which the *sapM-satS* gene arrangement is conserved (*Wattam et al., 2017*).

We constructed a Δ*satS* mutant of *M. tuberculosis* H37Rv to test if SatS is required for SapM secretion. The Δ*satS* mutant of *M. tuberculosis* did not exhibit an in vitro growth defect (*Figure 2—figure supplement 2A*). We monitored SapM secretion into culture media by immunoblotting culture filtrate proteins (CFPs) prepared from H37Rv, the Δ*secA2* mutant, the Δ*satS* mutant, and the Δ*satS* mutant complemented with $satS_{Mtb}$ using SapM antibodies. As expected, a SapM secretion defect was observed in the Δ*secA2* mutant. Even more striking was the SapM secretion defect of the Δ*satS* mutant, which was reproducibly more severe than the Δ*secA2* mutant (*Figure 2A*). This phenotype could be partially complemented with a $satS_{Mtb}$ plasmid that produced 26% of wild type levels of SatS (*Figure 2A* and *Figure 2—figure supplement 3A*). As controls, we immunoblotted the CFPs for detection of Mpt32, which is exported in a SecA2-independent manner and was not affected in the Δ*satS* mutant (*Figure 2A*), and also for the cytoplasmic SigA protein to rule out cell lysis contaminating the culture filtrates (*Figure 2—figure supplement 3B*). Since SapM is a phosphatase, we also quantified SapM secretion by measuring phosphatase activity in the culture filtrates, using p-Nitrophenyl Phosphate (PNPP) as a substrate. Consistent with the immunoblot data, there was significantly less phosphatase activity in the supernatant of a Δ*secA2* mutant compared to H37Rv, the Δ*satS* mutant exhibited an even more severe reduction in secreted phosphatase activity, and the Δ*satS* mutant phenotype could be complemented (*Figure 2B*). These results extend our identification of SatS as a SecA2 suppressor by revealing a role of SatS in the SecA2-dependent secretion of SapM by *M. tuberculosis*.

Even though *M. smegmatis* lacks a SapM orthologue, when we expressed *M. tuberculosis sapM* in *M. smegmatis*, SapM was also secreted in a SecA2 and SatS dependent manner (*Figure 2C*). Again, the SapM secretion defect of a Δ*satS* mutant was more severe than that of a Δ*secA2* mutant and this phenotype could be complemented (*Figure 2C*). As controls we immunoblotted for secreted Mpt32, which was not affected by the Δ*satS* mutation (*Figure 2C*), and for the cytoplasmic GroEL protein to rule out cell lysis (*Figure 2—figure supplement 3C*). This result indicates functional conservation of SatS in *M. smegmatis* and *M. tuberculosis*, and it indicates that the more amenable *M. smegmatis* is a valid model for studying SatS function.

To develop a higher throughput method for monitoring SatS and SecA2-dependent export, we established a whole cell assay for measuring secreted SapM phosphatase activity from *M. smegmatis* grown in 96 well plates. Importantly, this assay was specific for secreted SapM; it did not detect cytoplasmic SapM, as demonstrated by background levels of phosphatase activity of a *M. smegmatis* strain expressing non-exported cytoplasmic SapM lacking a signal sequence (Δss-SapM). In contrast, *M. smegmatis* expressing full length SapM preprotein, which is secreted, exhibited significantly greater activity (*Figure 2D*). When the Δ*secA2* mutant and the Δ*satS* mutant were tested in this whole cell phosphatase assay, the results confirmed the immunoblot data. Secreted phosphatase activity was reduced in both Δ*satS* and Δ*secA2* mutants, and the reduction was significantly more dramatic in the *satS* mutant (*Figure 2D*). The reduced activity of the Δ*satS* mutant could be complemented with either $SatS_{Mtb}$ or $SatS_{Msm}$ (*Figure 2D*).

## SatS effects on cellular levels of SapM

In addition to the reduced levels of secreted SapM, the total cellular (*i.e.* in whole cell lysate) and cytoplasmic levels of SapM were dramatically reduced in the *M. tuberculosis* Δ*satS* mutant compared to H37Rv, the Δ*secA2* mutant and the complemented Δ*satS* strains (*Figure 2E and F*). Notably, the reduction observed in the Δ*satS* mutant differed from a modest intracellular reduction of SapM in the Δ*secA2* mutant, and was specific to SapM as the levels of SigA were comparable across strains (*Figure 2E and F*). The same results were obtained with SapM-expressing *M. smegmatis* strains

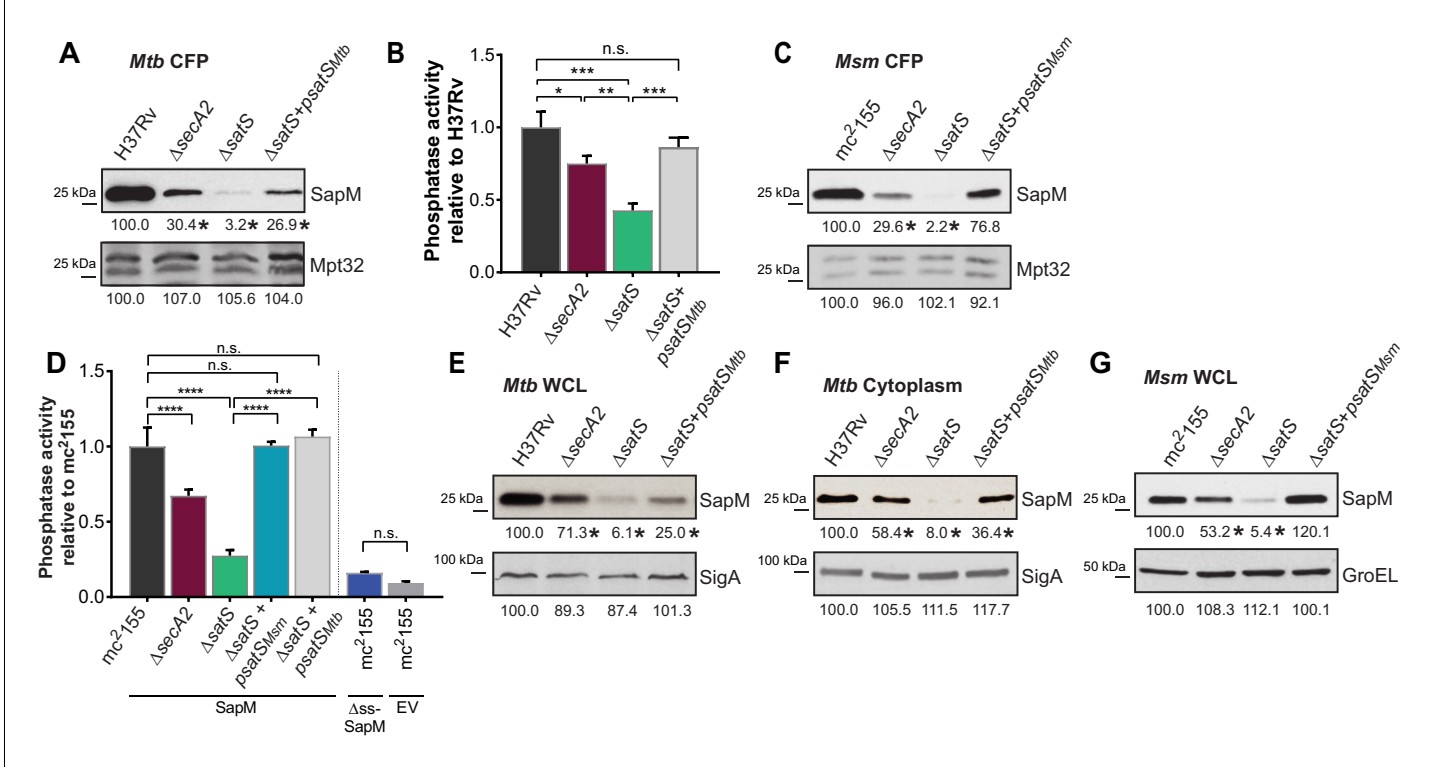

**Figure 2.** SatS is required for the export of SapM. (**A**) Equal protein from culture supernatants from *M. tuberculosis* H37Rv, Δ*secA2*, Δ*satS* and the complemented strain (Δ*satS* +*psatS*$_{Mtb}$) were immunoblotted for SapM and Mpt32. (**B**) Phosphatase activity in triplicate culture supernatant samples was examined by quantifying cleavage of pNPP. Rates of pNPP cleavage were normalized to H37Rv. (**C**) Equal protein from culture supernatants from *M. smegmatis* mc²155, Δ*secA2*, Δ*satS* and the complemented strain (Δ*satS* +*psatS*$_{Msm}$) were examined for levels of SapM and Mpt32 by Immunoblot (**D**) Whole cell phosphatase activity assay in *M. smegmatis*. All strains are expressing SapM, SapM lacking its signal sequence (Δss-SapM), or an empty vector as indicated. Triplicate wells containing $2 \times 10^5$ cells/well were grown in a 96 well plate for 24 hr at 37°C before measuring phosphatase activity by quantifying cleavage of pNPP. Rates were normalized to mc²155 +SapM. (**E**) Equal protein levels from whole cell lysates prepared from the same cultures as used for panel A or (**F**) the soluble, cytoplasmic fraction of *M. tuberculosis* H37Rv, Δ*secA2*, Δ*satS* and the complemented strain (Δ*satS* +*psatS*$_{Mtb}$) were immunoblotted for SapM and SigA. (**G**) Equal protein levels from whole cell lysates of prepared from the same cultures as used for panel C were immunoblotted for SapM and GroEL. Densitometry of blots from three experiments was performed (ImageJ). Percent difference of the mean intensity relative to wild-type is reported below each immunoblot. All data are representative of at least three independent experiments and all error bars represent standard deviation of the mean of three independent replicates for each strain. n.s. – no significant difference; *, p<0.05; **, p<0.01; ***, p<0.001; ****, p<0.0001 by ANOVA and Tukey's post hoc test.

DOI: https://doi.org/10.7554/eLife.40063.004

The following figure supplements are available for figure 2:

**Figure supplement 1.** *satS* and *sapM* are co-transcribed.

DOI: https://doi.org/10.7554/eLife.40063.005

**Figure supplement 2.** Characterization of the *M.tuberculosis* Δ*satS* mutant.

DOI: https://doi.org/10.7554/eLife.40063.006

**Figure supplement 3.** SatS$_{Mtb}$ complementation and lysis controls.

DOI: https://doi.org/10.7554/eLife.40063.007

**Figure supplement 4.** SatS does not affect *sapM* transcription or translation.

DOI: https://doi.org/10.7554/eLife.40063.008

(*Figure 2G*). In *M. smegmatis*, GroEL was used as a loading control and is comparable across strains (*Figure 2G*).

The reduced cellular and cytoplasmic levels of SapM in the Δ*satS* mutant might have reflected transcriptional or translational effects of SatS on *sapM*. Alternatively, SatS might act post-translationally to stabilize SapM prior to its export. Using quantitative Real-Time PCR (qRT-PCR) (*Figure 2—figure supplement 4A*) and a translational *sapM'-'lacZ* fusion (*Figure 2—figure supplement 4B*), we ruled out the possibilities of SatS functioning in transcription or translation. Thus, the effect of SatS

on SapM levels in the cytoplasm was post-translational, which leaves the most likely role for SatS being to protect SapM protein prior to export.

## Mce proteins exported by the SecA2 pathway require SatS

We also tested if SatS had an effect on additional SecA2 substrates. Multiple protein components of Mce transporters, which import lipids, depend on SecA2 to be exported to the cell wall (*Feltcher et al., 2015*). Immunoblot analysis of *M. tuberculosis* samples with Mce1A and Mce1E

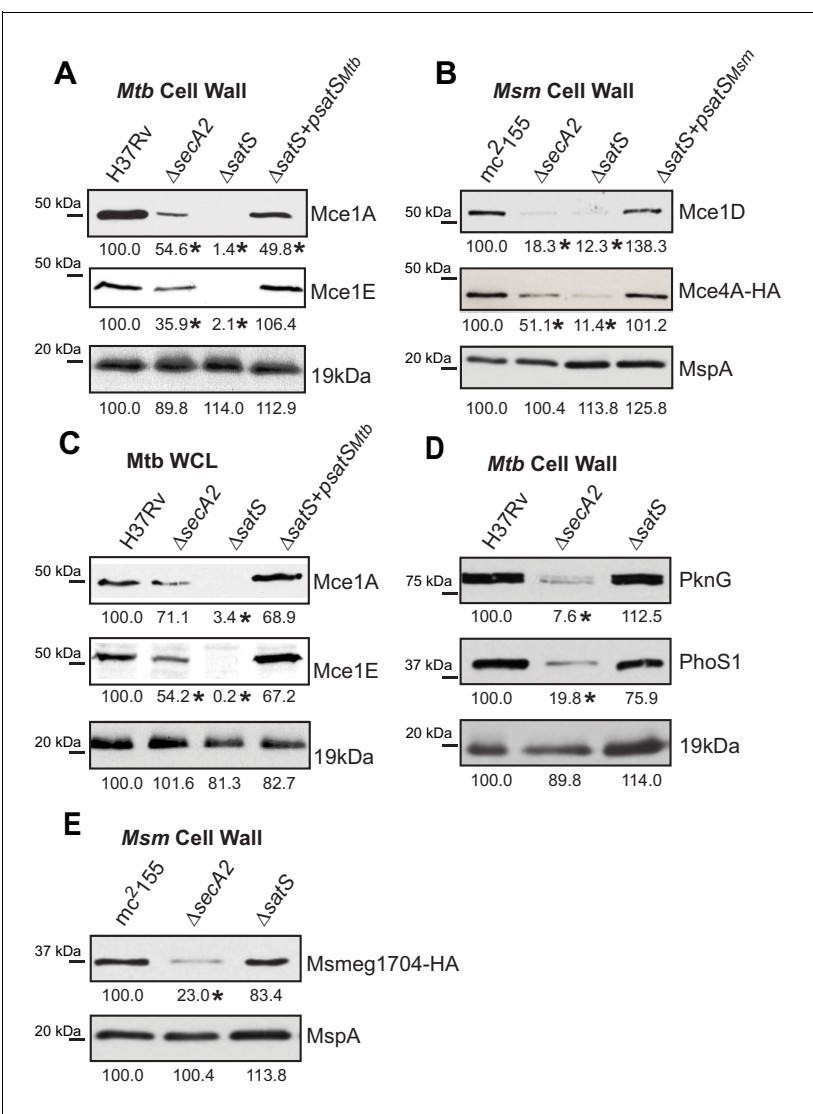

**Figure 3.** Mce proteins require SatS. (**A**) Equalized cell wall fractions of *M. tuberculosis* H37Rv, Δ*secA2*, Δ*satS* and complemented (Δ*satS* +*psatS*$_{Mtb}$) strains were analyzed by immunoblot using Mce1A, Mce1E, and 19 kDa antibodies to monitor differences in protein levels. (**B**) Equalized *M. smegmatis* mc[2]155, Δ*secA2*, Δ*satS*, and Δ*satS* +*psatS*$_{Msm}$ cell wall fractions were analyzed by immunoblot using Mce1D, HA (Mce4A-HA), and MspA antibodies. (**C**) Equalized protein from whole cell lysates of *M. tuberculosis* H37Rv, Δ*secA2*, Δ*satS* and the complemented strain (Δ*satS* +*psatS*$_{Mtb}$) were immunoblotted for Mce1A, Mce1E, and 19 kDa. (**D**) Equalized cell wall fractions of *M. tuberculosis* H37Rv, Δ*secA2*, and Δ*satS* strains were analyzed by immunoblot using PknG, PhoS1, and 19 kDa antibodies. (**E**) Equalized *M. smegmatis* mc[2]155, Δ*secA2*, and Δ*satS*, cell wall fractions were analyzed by immunoblot using HA (Msmeg1704-HA) and MspA antibodies. Densitometry of blots from three experiments was performed (ImageJ). Percent difference of the mean intensity relative to wild-type is reported below each immunoblot. *, p<0.05 by ANOVA and Tukey's post hoc test.

DOI: https://doi.org/10.7554/eLife.40063.009

antibodies revealed that the levels of Mce1A and 1E were reduced in cell wall of a Δ*satS* mutant (*Figure 3A*), with the defect in the Δ*satS* mutant again being more severe than the Δ*secA2* mutant. Mce importers are conserved in *M. smegmatis* and similar results were obtained upon immunoblotting *M. smegmatis* cell wall fractions with a Mce1D antibody (*Figure 3B*). The same effect was seen when a Mce4A$_{Msm}$-HA protein was expressed in *M. smegmatis* (*Figure 3B*). In contrast to these results, the level of the SecA2-independent 19 kDa lipoprotein in *M. tuberculosis* and MspA porin in *M. smegmatis* were unchanged in cell wall fractions of Δ*satS* mutants (*Figure 3A and B*). Like SapM, total cellular levels of Mce1A and Mce1E were also reduced in the *M. tuberculosis* Δ*satS* mutant compared to H37Rv, the Δ*secA2* mutant and the complemented Δ*satS* strain (*Figure 3C*).

We next tested whether SatS contributes to export of the SecA2-dependent protein kinase PknG and solute binding protein PhoS1 of *M. tuberculosis*, as well as the solute-binding protein Msmeg1704 of *M. smegmatis* (*Feltcher et al., 2013*; *Feltcher et al., 2015*; *van der Woude et al., 2014*). Immunoblotting of cell wall fractions confirmed that PknG, PhoS1, and Ms1704 depend on SecA2 for export; however, export of these proteins was not impaired in a Δ*satS* mutant (*Figure 3D and F*). These data demonstrate a level of specificity in the exported proteins affected by SatS. SatS affects multiple, but not all, of the proteins exported by the SecA2 pathway.

## SatS is required for *M. tuberculosis* growth in macrophages

The dramatic reductions in export of SapM and Mce proteins in the Δ*satS* mutant suggested that SatS is required for the pathogenesis of *M. tuberculosis*. SapM functions in limiting *M. tuberculosis* delivery to degradative lysosomes in macrophages while Mce proteins import lipids and thereby contribute to *M. tuberculosis* growth in macrophages and persistence in the host (*Vergne et al., 2005*; *Puri et al., 2013*; *Wilburn et al., 2018*). We tested a role for SatS during growth in macrophages by infecting murine bone marrow-derived macrophages with *M. tuberculosis* H37Rv, the Δ*secA2* mutant, Δ*satS* mutant, or Δ*satS* mutant complemented with *satS*$_{Mtb}$, and comparing intracellular growth over time. Compared to H37Rv, the Δ*satS* mutant demonstrated a significant defect in intracellular growth that was comparable to the previously demonstrated, attenuated phenotype of the Δ*secA2* mutant, and the Δ*satS* mutant phenotype could be complemented (*Figure 4*) (*Sullivan et al., 2012*; *Kurtz et al., 2006*). Thus, like SecA2, SatS plays an important role in enabling *M. tuberculosis* growth in macrophages even though only a subset of SecA2 substrates are affected by SatS.

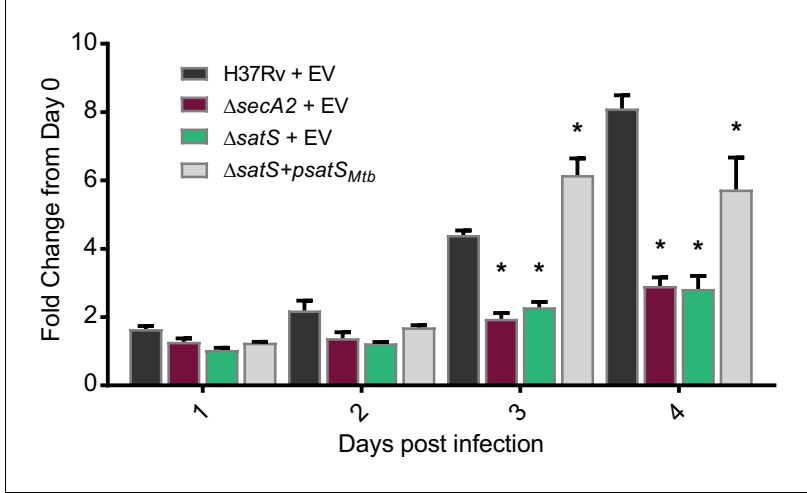

**Figure 4.** SatS contributes to *M. tuberculosis* growth in macrophages. Nonactivated BMDM were infected at an MOI of one with *M. tuberculosis* H37Rv + EV, Δ*secA2* + EV, Δ*satS* + EV, or Δ*satS* + psatS, and CFU burden was monitored over the course of a 4 day infection. The fold change in CFU over the course of the 4 day macrophage infection for each strain was calculated; the points represent means of triplicate wells, and the error bars represent standard deviations (SD). *, p<0.01; when compared to H37Rv by ANOVA and Tukey's post hoc test. Shown is a representative experiment of four independent experiments.
DOI: https://doi.org/10.7554/eLife.40063.010

## SatS and SapM interact

SatS has no sequence similarity to help predict function. However, the post-translational effect of SatS on cellular SapM levels was reminiscent of protein export chaperones of Type III and Type VII secretion systems (T3SS and T7SS), which stabilize their substrates in the cytoplasm and protect them from degradation prior to assisting in their secretion (*Thomas et al., 2012*; *Korotkova et al., 2014*). Thus, we considered the possibility that SatS is a protein export chaperone for specific substrates. To perform their functions, protein export chaperones interact with their substrates.

We tested whether SatS interacts with SapM in mycobacteria using co-immunoprecipitation. Immunoprecipitations were performed from *M. smegmatis* strains co-expressing C-terminally tagged SapM-FLAG and C-terminally tagged SatS-HA proteins. These epitope tags did not disrupt SapM or SatS functions (*Figure 5—figure supplement 1A and B*).

Reasoning that it may be easier to detect a SatS-SapM interaction when SapM export was compromised, we first performed co-immunoprecipitations in a $\Delta secA2$ mutant background. For these experiments we used a $\Delta secA2/\Delta satS$ double mutant expressing $SatS_{Mtb}$ ± HA tag and SapM-FLAG and immunoprecipitated from whole cell lysates using anti-HA agarose. The resulting immunoprecipitates were analyzed by immunoblotting with FLAG antibodies to detect SapM and SatS antibodies to detect SatS. SapM-FLAG was detected in the immunoprecipitates of the samples from the strain expressing $SatS_{Mtb}$-HA (*Figure 5A*) indicating SatS and SapM interact. As a control, SapM-FLAG was not recovered when the anti-HA immunoprecipitation was performed from a strain expressing untagged $SatS_{Mtb}$. Using high percentage (15%) SDS-PAGE, we detected two SapM-FLAG species: a ~ 31 kDa product corresponding to full length preprotein and a ~ 29 kDa product corresponding to the cleaved exported product. We confirmed the assignment of the smaller species as mature, cleaved SapM by immunoblotting lysate from a strain expressing Δss-SapM-FLAG (*Figure 5A*). It is striking that while the smaller exported species was more abundant in the input lysate, the full length preprotein SapM was the species that co-immunoprecipitated with SatS (*Figure 5A*). This is consistent with SatS interacting with SapM preprotein in the cytoplasm, prior to its export and signal sequence cleavage. We investigated whether the signal sequence of SapM is required for the SatS-SapM interaction by immunoprecipitating from a strain co-expressing SatS-HA and Δss-SapM-FLAG. SatS-HA and Δss-SapM-FLAG co-immunoprecipitated indicating that the signal sequence is not required for the SatS-SapM interaction (*Figure 5A*).

We were also able to co-immunoprecipitate SapM-FLAG with SatS-HA from cell lysates of a *M. smegmatis* Δ*satS* strain expressing the same constructs (*i.e.* a *secA2* wild-type background), although there was reproducibly less SapM-FLAG recovered when export was not inhibited (*Figure 5B*). Once again, the SapM preprotein species preferentially co-immunoprecipitated with SatS. To address the specificity of SatS interacting with SapM, we also immunoblotted SatS immunoprecipitates with antibody to MspA, which is a cell wall porin that is exported in a SecA2 and SatS-independent manner (*Wolschendorf et al., 2007*; *Feltcher et al., 2013*) (*Figure 2F*). MspA did not co-immunoprecipitate with SatS (*Figure 5B*).

The interaction between the preprotein species of SapM and SatS implied that SatS is a cytoplasmic protein. Using an antibody raised against SatS, we confirmed that in both *M. tuberculosis* and *M. smegmatis* SatS is cytoplasmic (*Figure 2—figure supplement 2B* and *Figure 1—figure supplement 1B*). Interestingly, SatS in *M. tuberculosis* and *M. smegmatis* migrated on SDS-PAGE at ~65 kDa rather than at its predicted molecular weight of 46 kDa. SatS purified from *Escherichia coli* also ran at 65 kDa (data not shown).

## SatS functions prior to SecA2

If SatS functions as a chaperone for preproteins exported by the SecA2 pathway, we predicted its role should come before the role of SecA2 in exporting SapM across the membrane. To test this order of events, we constructed a *M. smegmatis* Δ*secA2*/Δ*satS* double mutant expressing SapM-FLAG and compared the cellular and secreted levels of SapM of the double mutant to single Δ*secA2* or Δ*satS* mutants. If SatS acts prior to SecA2 in exporting SapM, the Δ*satS* mutation should be epistatic to the Δ*secA2* mutation, which proved to be the case. The Δ*secA2*/Δ*satS* double mutant exhibited the equivalent dramatic reduction in cellular and secreted levels of SapM as exhibited by the Δ*satS* mutant (*Figures 6A, B and C*). Further, there was no additive effect evident on the SapM secretion defect in the Δ*secA2*/Δ*satS* double mutant compared to the Δ*satS* mutant (*Figure 6B*).

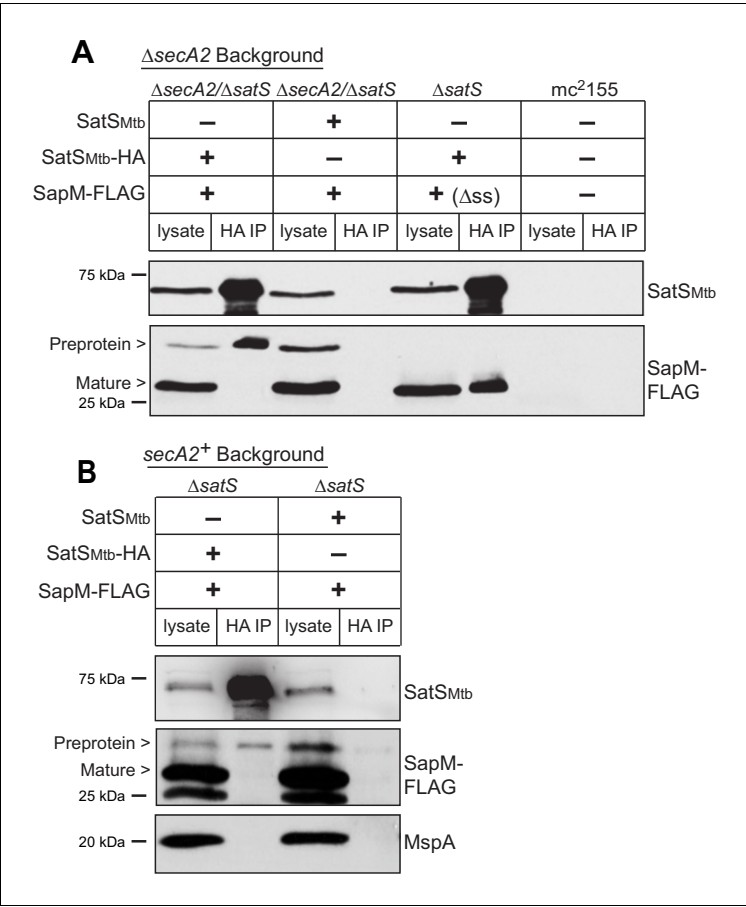

**Figure 5.** SatS and SapM interact. (**A**) Lysate from *M. smegmatis* Δ*secA2*/Δ*satS* expressing SapM-FLAG and either SatS*Mtb*-HA or SatS*Mtb* without a tag, Δ*satS* expressing Δss-SapM-FLAG and SatS*Mtb*-HA, or mc²155 with two empty vectors (as shown above the blot) were used for co-immunoprecipitation using anti-HA conjugated beads. Lysates (left) and immunoprecipitations (right) for each strain were probed with SatS antibody and FLAG antibody for SapM. Two different sizes of SapM-FLAG corresponding to the full-length (signal sequence-containing) and mature (cleaved signal sequence) species were detected. (**B**) Lysate from *M. smegmatis* Δ*satS* expressing SapM-FLAG and either SatS*Mtb*-HA or SatS*Mtb* without a tag were used for co-immunoprecipitation using anti-HA conjugated beads. Lysates (left) and co-immunoprecipitations (right) for each strain were probed with SatS antibody, FLAG antibody, and MspA antibody. All data are representative of at least three independent experiments.

DOI: https://doi.org/10.7554/eLife.40063.011

The following figure supplement is available for figure 5:

**Figure supplement 1.** Epitope tags do not disrupt SapM or SatS functions.
DOI: https://doi.org/10.7554/eLife.40063.012

## SatS behaves as a chaperone to prevent SapM aggregation

The data thus far were consistent with the hypothesis that SatS functions as a chaperone for a subset of SecA2 dependent substrates. The hallmark of a chaperone is that it binds to unfolded or misfolded regions of proteins to prevent inappropriate interactions, such as aggregation (*Ellis, 1997*). To obtain more direct evidence for chaperone activity of SatS, we purified SatS*Mtb* and preSapM-His (full length SapM preprotein containing the signal sequence) from *E. coli* and tested the ability of SatS to prevent aggregation in vitro of preSapM-His. preSapM-His was solubilized from inclusion bodies using 8 M urea, rapidly diluted into refolding buffer (150 fold), and its aggregation was followed by change in light scattering at 350 nm. In the absence of SatS, dilution of denatured preSapM-His rapidly led to the formation of light scattering aggregates (*Figure 7*). However, inclusion of SatS in the dilution buffer prevented preSapM-His aggregation in a dose-dependent manner

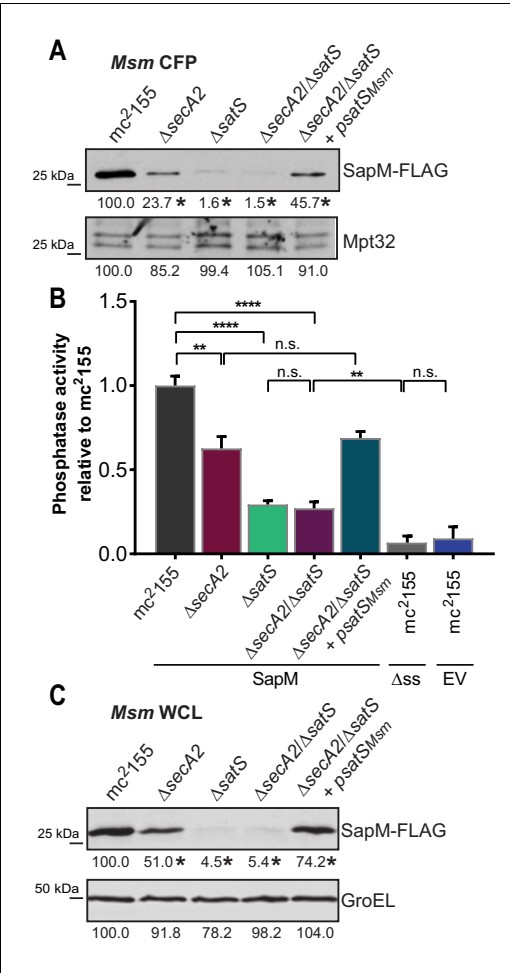

**Figure 6.** SatS functions prior to SecA2. (**A**) Equal protein from culture supernatants (CFP) from *M. smegmatis* mc²155, Δ*secA2*, Δ*satS*, the Δ*secA2/Δ satS* double mutant, and Δ*secA2/ΔsatS* expressing wild-type SatS (Δ*secA2/ΔsatS +psatS_Msm*) were examined for levels of SapM-FLAG and Mpt32, by Immunoblot. (**B**) Whole cell phosphatase activity assay using the above *M. smegmatis* strains. All strains are expressing SapM, SapM lacking its signal sequence (Δss-SapM), or an empty vector as indicated. Rates were normalized as described above. (**C**) Whole cell lysates from the above *M. smegmatis* strains were examined for levels of SapM-FLAG and GroEL by Immunoblot. For panels A and C, densitometry of blots from three experiments was performed (ImageJ). Percent difference of the mean intensity relative to wild-type is reported below each immunoblot. All data are representative of at least three independent experiments and all error bars represent standard deviation of the mean of three independent replicates for each strain. n.s. – no significant difference; **, p<0.01; ****, p<0.0001 by ANOVA and Tukey's post hoc test.

DOI: https://doi.org/10.7554/eLife.40063.013

(*Figure 7*). As controls, BSA and lysozyme did not reduce preSapM-His aggregation (*Figure 7*). In fact, BSA or lysozyme modestly had the opposite effect of increasing the light scattering signal. A SatS:preSapM-His molar ratio of 2.5:1 was sufficient to completely ablate preSapM-His aggregation and even a 0.5:1 ratio was sufficient to reduce aggregation by 33%. The data from this in vitro anti-aggregation assay provide strong support for SatS acting as a chaperone and for a direct interaction between SatS and SapM preprotein.

## SatS has a new fold and hydrophobic grooves that share similarity with the preprotein binding sites of the SecB chaperone

Although the amino acid sequence of SatS bears no similarity to any known chaperones, the data above support a role for SatS as a protein export chaperone. To gain further insight into SatS function, we collected diffraction quality crystals and determined the crystal structure of SatS to 2.3 Å. Upon inspection, the electron density map only corresponded to the last 185 amino acids (L237-E420) C-domain of the SatS sequence (SatS_C). The molecular weight of the SatS_C crystal was ~25 kDa as determined by SDS-PAGE, indicating that the protein underwent in situ proteolysis in the crystallization buffer. Further investigation revealed striking similarity between the experimentally derived SatS_C secondary structure and the predicted secondary structure of the first ~180 amino acids of the N-domain of SatS (SatS_N) (*Figure 8—figure supplement 1A and B*). The SatS_C and SatS_N domains are also similar in size with 41% sequence similarity at the amino acid level (*Figure 8—figure supplement 1A and B*). This raised the possibility that SatS is comprised of two similar domains with an intervening ~60 amino acids that is predicted to be a flexible, disordered linker (*Figure 8—figure supplement 1A*). Subsequently, constructs expressing SatS_C or SatS_N in *E. coli* were used to purify the individual domains to homogeneity for crystallization trials. Only SatS_C yielded diffraction quality crystals diffracting to 1.4 Å resolution (*Figure 8A*).

SatS_C displays α/β secondary structure comprised of a mostly parallel, four stranded β-sheet core, flanked by seven α-helices (*Figure 8A*). The structure revealed a new fold sharing no similarities with any previously solved protein structure in the PDB based on DELTA-BLAST and VAST similarity searches (*Boratyn et al., 2012*;

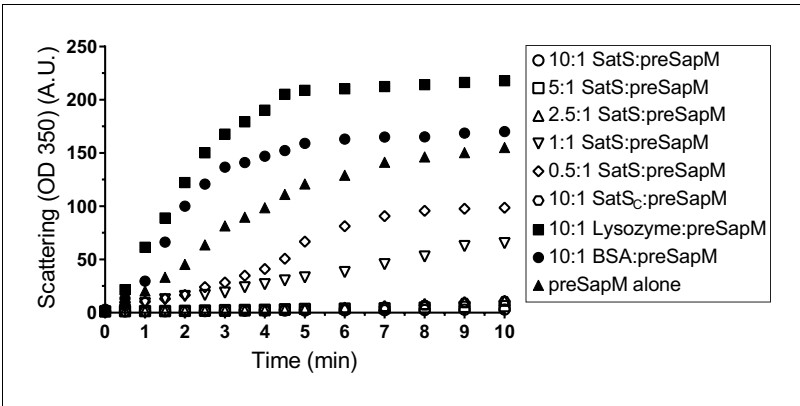

**Figure 7.** SatS and SatS$_C$ prevent aggregation of SapM *in vitro*. Denatured SapM-His was diluted 150 fold to a final concentration of 1 μM in 40 mM HEPES, 100 mM NaCl, pH 7.4. SapM-His aggregation was monitored by light scattered (350 nm) at 25°C in the presence or absence of SatS/SatS$_C$ or, as controls, lysozyme or BSA. A molar ratio of 2.5:1 of SatS:SapM-His could prevent SapM-His aggregation and aggregation was significantly reduced using a molar ratio of 0.5:1.

DOI: https://doi.org/10.7554/eLife.40063.014

*Madej et al., 2014*). Although the overall polypeptide fold was not similar to known proteins, the surface of SatS$_C$ had pronounced electronegative charge potential that is comparable to many export chaperones, including SecB (*Francis, 2010*). Furthermore, the SatS$_C$ structure featured two surface localized hydrophobic grooves, mapped by the Kyte-Doolittle hydrophobicity scale (*Kyte and Doolittle, 1982*) (*Figure 8A*). These grooves bore similarity to the hydrophobic grooves on SecB (*Figure 8B*) that serve as primary and secondary client binding sites to regions of unfolded preproteins (*Xu et al., 2000*). The proximity of the smaller hydrophobic groove in SatS$_C$ (Site 2) to the larger groove in SatS$_C$ (Site 1) as well as their amino acid composition (aromatic and bulky side chains) resembled the arrangement and composition of the client binding sites of SecB. The larger of the two hydrophobic grooves in SatS$_C$ (Site 1) was comparable in size to the ~60 Å long, main binding site in SecB. Because of the similarities between SecB and SatS$_C$, we speculated that SatS$_C$ may be sufficient to perform SatS chaperone functions. In fact, when we tested SatS$_C$ for chaperone activity in the in vitro anti-aggregation assay, SatS$_C$ alone was sufficient to ablate preSapM-HIS aggregation comparable to full length SatS (*Figure 7*).

## SatS has at least two separable roles in protein export

The majority of *satS* suppressor mutations were expected to behave like *satS* null mutations (*Figure 1B*). However, one mutation (3S) that caused a single amino acid substitution (G134D) produced wild-type levels of full length SatS protein when compared to wild-type SatS expressed from the same vector backbone (*Figure 9C*). Using this expression plasmid, we tested the importance of the G134 residue, which is ubiquitous in SatS homologs in mycobacteria.

We first tested if *satS G134D* could complement the SapM-FLAG secretion defect of the *M. smegmatis ΔsatS* mutant by immunoblotting culture filtrates. Since *satS G134D* behaved like the *ΔsatS* null mutant in suppressing *secA2 K129R*, we predicted that *satS G134D* would fail to complement the SapM secretion defect of the *ΔsatS* mutant. Along these lines, *satS G134D* exhibited a SapM secretion defect. However, with either *satS$_{Msmeg}$ G134D* or *satS$_{Mtb}$ G134D* the SapM secretion defect was comparable to the level of the secretion defect of the *ΔsecA2* mutant not a *ΔsatS* mutant in the immunoblot and whole cell secreted phosphatase activity assays (*Figure 9A and B*).

We also evaluated the effect of SatS G134D on cellular SapM levels. To our surprise, *satS G134D* did not behave like a *ΔsatS* mutant. Rather, it fully complemented the dramatic *ΔsatS* reduction in SapM levels seen in whole cell lysates (*Figure 9C*). Furthermore, we were able to co-immunoprecipitate SatS G134D-HA and SapM-FLAG preprotein (*Figure 9D*) indicating that SatS G134D retains the ability to interact with SapM-FLAG. Our discovery that SatS G134D still binds SapM preprotein and maintains cellular levels of SapM, yet *satS G134D* exhibits a defect in SapM secretion equivalent to

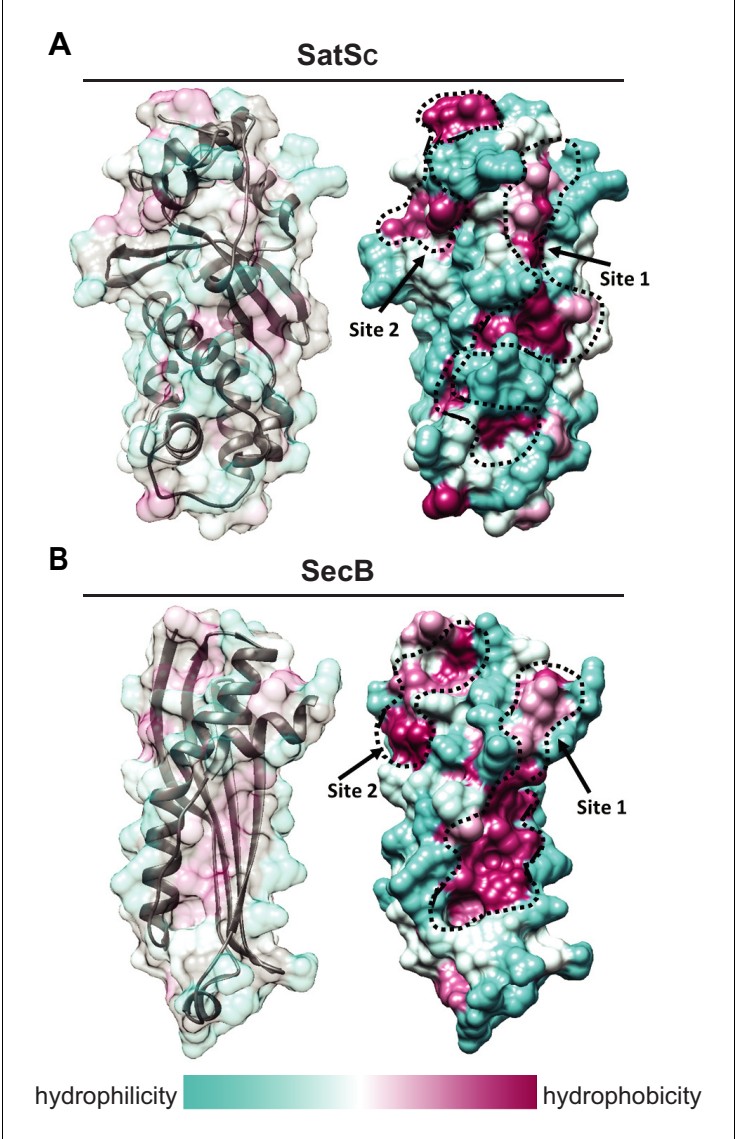

**Figure 8.** SatS has a new fold and hydrophobic grooves that share similarity with the preprotein binding sites of the SecB chaperone. (A) The overall secondary structure of $SatS_C$. The hydrophilicity of $SatS_C$ is a colored gradient from cyan (hydrophilic) to maroon (hydrophobic). $SatS_C$ exposes ~2,900 $Å^2$ of hydrophobic surface. The predicted primary and secondary polypeptide binding site(s) are delineated. (B) The overall secondary structure of SecB monomer (PDB ID:1QYN) (*Dekker et al., 2003*). The hydrophilicity of SecB is a colored gradient from cyan (hydrophilic) to maroon (hydrophobic). The primary and secondary client binding site(s) are delineated. Each SecB monomer exposes ~1,900 $Å^2$ of hydrophobic surface for client protein interactions (*Huang et al., 2016*). Molecular graphics and analyses were performed with the UCSF Chimera package (*Petersen et al., 2011*).
DOI: https://doi.org/10.7554/eLife.40063.015

The following source data and figure supplement are available for figure 8:

**Source data 1.** $SatS_C$ X-ray Structure Validation Details.
DOI: https://doi.org/10.7554/eLife.40063.017

**Figure supplement 1.** $SatS_C$ secondary structure and the predicted secondary structure of $SatS_N$.
DOI: https://doi.org/10.7554/eLife.40063.016

that of a *ΔsecA2* mutant, indicates that SatS has more than one role in SapM secretion by the SecA2 pathway. Moreover, these multiple functions of SatS in export can be uncoupled.

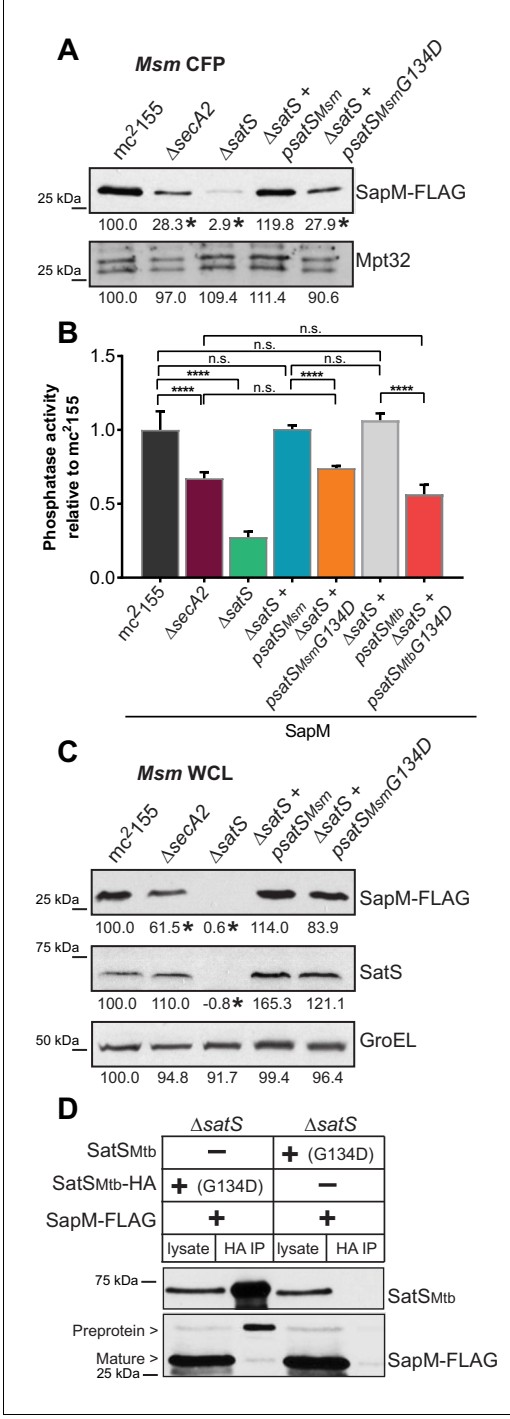

**Figure 9.** SatS has at least two separable roles in protein export. (**A**) Equal protein from culture supernatants (CFP) from *M. smegmatis* mc$^2$155, Δ*secA2*, Δ*satS* and the Δ*satS* mutant expressing either wild-type SatS (Δ*satS +psatS$_{Msm}$*) or SatS G134D (Δ*satS +psatS$_{Msm}$G134D*) were examined for levels of SapM-FLAG and Mpt32 by Immunoblot. (**B**) Whole cell phosphatase activity assay of *M. smegmatis* strains expressing SapM. Rates were normalized as described above. (**C**) Equal protein from whole cell lysates (WCL) *Figure 9 continued on next page*

## Discussion

As with all bacterial pathogens, the protein export pathways of *M. tuberculosis* are critical to virulence. Here, we identified SatS, a previously uncharacterized protein of unknown function, as a new protein export factor with a role in intracellular growth of *M. tuberculosis*. We further discovered multiple properties of SatS that indicate a function as a protein export chaperone. As the amino acid sequence of SatS bears no similarity to chaperones and the structure of the SatS$_C$ domain reveals a new fold, SatS appears to represent a new type of protein export chaperone.

## Suppressor analysis led to the identification of SatS

Suppressor analysis is a classic approach for identifying genes in pathways, and it was used extensively in early studies of the general Sec pathway in *E. coli* (*Bieker-Brady and Silhavy, 1992*; *Flower et al., 1995*). Here, we carried out a suppressor screen using *secA2 K129R*, which encodes a variant of SecA2 that is unable to hydrolyze ATP (*Rigel et al., 2009*). Past studies lead to a model where SecA2 K129R is locked in a nonfunctional complex with SecY while attempting to export its substrates (*Ligon et al., 2013*). As a result, SecA2 K129R is trapped at the membrane and SecY proteins are degraded (*Ligon et al., 2013*). Our discovery that loss-of-function *satS* mutations suppress *secA2 K129R* phenotypes suggests that SatS is required for the detrimental interaction of SecA2 K129R with SecYEG to occur. In fact, deletion of *satS* significantly reversed SecA2 K129R retention at the membrane and the associated SecY degradation, which is consistent with avoidance of the interaction. By extension, these results support a role for SatS in enabling wild-type SecA2 to interact with the SecYEG channel.

One possibility for how SatS promotes SecA2 interactions with the SecYEG channel is that SatS is a core component of a SecA2-specific export apparatus with a function mediating the interaction between SecA2 and SecYEG. However, if SatS were to function this way, we would expect all SecA2-dependent substrates would require SatS for export, which was not the case. An alternate possibility is that in order for SecA2 to be delivered to or engage the SecYEG channel it must first be bound to a substrate in a translocation competent state and that SatS functions as a protein export chaperone that facilitates this SecA2-substrate interaction. We favor this role for SatS as it would not only help explain why

*Figure 9 continued*

from *M. smegmatis* strains described above were examined for levels of SapM-FLAG, SatS and GroEL by Immunoblot. For panels A and C, densitometry of blots from three experiments was performed (ImageJ). Percent difference of the mean intensity relative to wild-type is reported below each immunoblot. (D) Lysate from *M. smegmatis* Δ*satS* expressing SapM-FLAG and either SatS*Mtb*G134D-HA or SatS*Mtb*G134D without a tag were used for co-immunoprecipitation using anti-HA conjugated beads. Lysates (left) and co-immunoprecipitations (right) for each strain were probed with SatS antibody and FLAG antibody. All data are representative of at least three independent experiments and all error bars represent standard deviation of the mean of three independent replicates for each strain. n.s. – no significant difference; ****, p<0.0001 by ANOVA and Tukey's post hoc test.
DOI: https://doi.org/10.7554/eLife.40063.018

phenotypes of *secA2 K129R* depend on the presence of SatS but it is also consistent with our identification of an interaction between SatS and SapM and the chaperone activities of SatS. However, a question raised by this model is how elimination of SatS suppresses *secA2 K129R* if there are also SatS-independent proteins that could interact with SecA2 K129R. The answer may be that the threshold for phenotypic suppression does not require all SecA2 K129R to be diverted from the SecYEG channel.

## SatS as a protein export chaperone

Molecular chaperones are defined by their ability to transiently bind unfolded regions of proteins and, thereby, protect them from inappropriate interactions, such as aggregation, incorrect/premature folding or degradation (*Ellis, 1997*). Chaperones are a common component of protein export systems, with SecB of the general Sec pathway in Gram-negative bacteria and Type III Secretion System Chaperones (T3SCs) being examples. In mycobacteria, EspG proteins of Type VII Secretion Systems are the only protein export chaperones identified so far (*Daleke et al., 2012*; *Ekiert and Cox, 2014*). As a subset of molecular chaperones, protein export chaperones have additional functions in export, such as targeting substrates to export machinery. Although there is a notable lack of amino acid and structural similarity between different types of protein export chaperones, commonalities exist. Protein export chaperones are all highly acidic (pI <5.0) proteins that transiently interact with their substrates in the cytoplasm and remain in the cytoplasm when the substrate is exported (*Randall and Hardy, 2002*; *Thomas et al., 2012*; *Ekiert and Cox, 2014*). Additionally, a hallmark of a protein export chaperone is that its role is limited to a subset of the proteins exported by a given system (*Collier et al., 1990*; *Thomas et al., 2005*; *Daleke et al., 2012*). Finally, in some cases, the genes encoding the chaperone and substrate are co-expressed and in an operon (*Parsot et al., 2003*; *Daleke et al., 2012*).

SatS has many features of a protein export chaperone. SatS is a highly acidic (pI 3.83), cytoplasmic protein with a role promoting export of a subset of the proteins exported by the SecA2 pathway. Further, the *satS* and *sapM* genes are co-transcribed in an operon and we obtained evidence of a SatS:SapM interaction occurring in mycobacteria. SatS preferentially interacted with the full-length preprotein of SapM indicating that the interaction occurs in the cytoplasm prior to SapM export. However, like other protein export chaperones (T3SCs, SecB, and EspG₅) (*Stebbins and Galán, 2001*; *Huang et al., 2016*; *Ekiert and Cox, 2014*) where binding occurs in regions of the mature domain of the substrate, the signal sequence was not required for the SatS-SapM interaction. The in vitro anti-aggregation effect of SatS on SapM preprotein provided the most direct proof of a SatS:SapM interaction and a chaperone function for SatS. Finally, in the absence of SatS, the level of SapM in the cytoplasm was dramatically reduced. This effect of SatS on intracellular SapM levels is post-translational and is also reminiscent of effects of T3SCs and EspG chaperones (*Thomas et al., 2005*; *Korotkova et al., 2014*).

In comparison to the dramatic reduction in intracellular SapM in the Δ*satS* mutant, the Δ*secA2* mutant exhibited only a modest effect on intracellular SapM levels. This difference in intracellular SapM levels translates to the more severe secretion defect of the Δ*satS* mutant versus the Δ*secA2* mutant. Export defects of mycobacterial Δ*secA2* mutants are never 100% (i.e. residual export is observed in Δ*secA2* mutants) (*Braunstein et al., 2001*; *van der Woude et al., 2014*; *Feltcher et al., 2015*; *Zulauf et al., 2018*). The pathway responsible for the residual export in a Δ*secA2* mutant remains unknown although the general Sec pathway involving SecA1 is an attractive candidate. Thus, it is possible that SatS also works with SecA1 and the general Sec pathway, at least when SecA2 is absent. Moreover, we cannot rule out the possibility that there are SatS substrates that are

exported in a completely SecA2-independent manner. Additional studies will be required to address these unknowns.

Because of the dramatically reduced levels of SapM in the whole cell lysate of the Δ*satS* mutant, it was not immediately clear if the role of SatS in SapM secretion was solely to maintain intracellular levels of SapM preprotein or if SatS had additional roles. By evaluating the *satS G134D* mutant, we revealed the existence of at least one additional role for SatS in promoting SapM secretion. In the *satS G134D* mutant, intracellular SapM was maintained at wild-type levels; yet, there remained a SapM secretion defect. It is noteworthy that the SapM secretion defect of the *satS G134D* mutant was on the order of a Δ*secA2* mutant, which is consistent with SatS G134 working in the SecA2 pathway. Future studies should address this second function, which could be a role for SatS in targeting substrates to the SecA2 pathway and/or in maintaining SapM in an unfolded state for protein translocation across SecYEG.

## The SatS structure defines a new fold with hydrophobic grooves typical of substrate binding sites

Although we set out to solve the structure of SatS in its entirety, we were only able to obtain structural information for the C-terminal half of the protein (SatS_C), which arose during crystallization. However, the primary sequence and secondary structure similarity between the N-terminal and C-terminal halves of SatS raise the possibility of SatS being comprised of tandem SatS_C-like domains. Investigation of the SatS_C structure revealed a large network of negatively charged amino acids surrounding two surface exposed hydrophobic grooves, which are similar in arrangement, shape and size to the hydrophobic client binding sites of a SecB monomer (*Huang et al., 2016*). In the solution structure of SecB in complex with a preprotein, the unfolded preprotein wraps around the SecB tetramer through interactions with the hydrophobic client binding sites. This binding architecture helps explain the means by which SecB maintains Sec preproteins in an unfolded state, as is required for their transport through the SecYEG channel (*Tsirigotaki et al., 2017*). The similarity in hydrophobic grooves in SatS and SecB is intriguing since SatS works with the SecA2 pathway, which also uses the SecYEG channel. Moreover, these similarities suggest that the hydrophobic grooves in SatS may serve as similar substrate binding sites. In fact, in the anti-aggregation assay the SatS_C domain was sufficient for preventing SapM preprotein aggregation, indicating that SatS_C is capable of directly interacting with SapM preprotein. Mycobacteria lack a canonical SecB protein export chaperone, although in *M. tuberculosis* there is a SecB-like protein that functions as a chaperone for a toxin-antitoxin system (*Bordes et al., 2011*). Thus, even though SecB and SatS are not evolutionarily conserved, it is interesting to speculate a SecB-like function for SatS. SatS may be an adaptation for export of specific proteins by Actinomycetales, since SatS orthologs are not found outside of this order.

## SatS is required for growth of *M. tuberculosis* in macrophages

Prior TraSH/Tnseq analyses using pooled libraries of transposon mutants predicted SatS to be required during murine and macrophage infections (*Rengarajan et al., 2005*; *Zhang et al., 2013*); however, this prediction had never been validated. Here, using a Δ*satS* mutant and a complemented strain, we directly demonstrated a role for SatS in *M. tuberculosis* growth in macrophages. These data argue for an important role of SatS and its specific substrates in pathogenesis. Given that only a subset of SecA2 substrates are affected by SatS, future studies should include investigation of SatS substrates and their contribution to pathogenesis. Since our approach for identifying SatS substrates was not exhaustive, there may also exist SatS-dependent proteins that remain to be identified.

## Conclusion

By way of a genetic screen in *M. smegmatis*, we identified a new protein SatS with roles in protein export in *M. tuberculosis*. This work not only expands our understanding of the specialized SecA2 protein export pathway of mycobacteria but it provides important functional information for a previously uncharacterized *M. tuberculosis* protein that contributes to pathogenesis. Further, by assigning a chaperone function to SatS, our studies expand our appreciation of the diversity of chaperones in biological systems. Although chaperones have common functions, substantial structural diversity

exists among these proteins, which is further highlighted by the new fold revealed in the structure of SatS_C.

# Materials and methods

| Reagent type (species) or resource | Designation | Source or reference | Identifiers | Additional information |
|---|---|---|---|---|
| Strain, *Mycobacterium tuberculosis* | MBTB508; (WT Mtb) | *Zulauf et al., 2018* | | *M. tuberculosis* wild-type H37Rv + Ev (pMV306.kan) |
| Strain, *M. tuberculosis* | MBTB443 | *Zulauf et al., 2018* | | ΔsecA2 + Ev (pMV306.kan) |
| Strain, *M. tuberculosis* | MBTB512 | this paper | | ΔsatS + Ev (pMV306.kan) |
| Strain, *M. tuberculosis* | MBTB513 | this paper | | ΔsatS + psatS (pBM13) |
| Strain, *Mycobacterium smegmatis* | mc²155; (WT Msm) | *Snapper et al., 1990* | | *M. smegmatis* wild-type (WT) |
| Strain, *M. smegmatis* | NR116 | *Rigel et al., 2009* | | ΔsecA2 |
| Strain, *M. smegmatis* | BAF1 | this paper | | ΔsecA2/ΔsatS |
| Strain, *M. smegmatis* | BM10 | this paper | | ΔsatS |
| Recombinant DNA reagent (primers) | satSMtb US flank F | this paper | GCGGTACCGCCGTG GGTCAACTTCAGTAAC | Contains engineered KpnI site, used to amplify US flank for pSM42 |
| Recombinant DNA reagent (primers) | satSMtb US flank R | this paper | GCGTCTAGAGGTG CTGATGATCTCGTCGATG | Contains engineered XbaI site, used to amplify US flank for pSM42 |
| Recombinant DNA reagent (primers) | satSMtb DS flank F | this paper | GCGAAGCTTATGATC GACCGATCTTCCTG | Contains engineered HindIII site, used to amplify DS flank for pSM42 |
| Recombinant DNA reagent (primers) | satSMtb DS flank R | this paper | GCGACTAGTCGGGC TGTTTTCTACGTTGT | Contains engineered SpeI site, used to amplify DS flank for pSM42 |
| Recombinant DNA reagent (primers) | satSMsm US flank F | this paper | AACATATGCGCAACTG GGTGTGCCGTATCACTG | Contains engineered NdeI site, used to amplify US flank for pLL50 |
| Recombinant DNA reagent (primers) | satSMsm US flank R | this paper | AAGCTAGCAGCAGCCAT GCGGCACAGCCTAAC | Contains engineered NheI site, used to amplify US flank for pLL50 |
| Recombinant DNA reagent (primers) | satSMsm DS flank F | this paper | ATGCTAGCTCCCGGC TCCGTCAGGAGTAGCG | Contains engineered NheI site, used to amplify DS flank for pLL50 |
| Recombinant DNA reagent (primers) | satSMsm DS flank R | this paper | AACATATGAGCCACCCG GCGAAATTGAAGCCAC | Contains engineered NdeI site, used to amplify DS flank for pLL50 |
| Recombinant DNA reagent (primers) | 1684-F-Native | this paper | AGTTAATTAACGTGTG CTCGACGGCCTGGTTGCC | Contains engineered PacI site, used to construct satSMsm plasmids pBM4, pBM22, and pBM23 |
| Recombinant DNA reagent (primers) | 1684 R | this paper | AATGGCCACTACTCCTG ACGGAGCCGGGACTCCAC | Contains engineered BalI site, used to construct satSMsm plasmids pBM4 and pBM22 |

*Continued on next page*

*Continued*

| Reagent type (species) or resource | Designation | Source or reference | Identifiers | Additional information |
|---|---|---|---|---|
| Recombinant DNA reagent (primers) | 1684 R-HA | this paper | ATTGGCCATCAG GCGTAGTCCGGCAC GTCGTACGGG TACTCCTGACGGAGC CGGGACTCCAC | Contains engineered HA tag and BamHI site, used to construct satSMsm-HA plasmid pBM23 |
| Recombinant DNA reagent (primers) | Rv3311-F | this paper | AATGGCCACTGACC TCGTACCCATCCGCTTGAG | Contains engineered MscI site, used to construct satSMtb plasmids pBM13, pBM60, and pBM80 |
| Recombinant DNA reagent (primers) | Rv3311-R | this paper | AATGGCCACTAGC CTTCGCCGGCTGAC | Contains engineered MscI site, used to construct satSMtb plasmid pBM80 |
| Recombinant DNA reagent (primers) | Rv3311-HA-R | this paper | TTAAGCTTCGCGCC TGAGCCGCGACTCC | Contains engineered HindIII site, used to construct satSMtb plasmids pBM13 and pBM60 |
| Recombinant DNA reagent (primers) | Rv3311-G134D-F | this paper | GCCCAGGATGGGAT TGTCGTTGAAGAACTTCGA | Used for site directed mutagenesis on pBM80 to generate pBM87 |
| Recombinant DNA reagent (primers) | Rv3311-G134D-R | this paper | TCGAAGTTCTTCAAC GACAATCCCATCCTGGGC | Used for site directed mutagenesis on pBM80 to generate pBM87 |
| Recombinant DNA reagent (primers) | SapM-F | this paper | TGGCCAACCGCG GAATCCAGGCTCTC | Contains engineered MscI site, used to construct sapM plasmids pJTS130, and pBM56 |
| Recombinant DNA reagent (primers) | Dss-SapM-F | this paper | TGGCCAAGACCT TCGCGCACGTGG | Contains engineered MscI site, used to construct sapM plasmids pJTS132 and pBM61 |
| Recombinant DNA reagent (primers) | SapM-R | this paper | AAGCTTCCATGCG GCACAGAATAGCGAC | Contains engineered HindIII site, used to construct sapM plasmids pJTS130 and pJTS132 |
| Recombinant DNA reagent (primers) | SapM-FLAG-R | this paper | ACTAAGCTTTCACTT GTCGTCGTCGT CCTTGTAGT CCGTATACGAGCCGCCG TCGCCCCAAATATCG | Contains engineered linker-FLAG and HindIII site, used to construct sapM plasmids pBM56 and pBM61 |
| Recombinant DNA reagent (primers) | SapM-Promoter F | this paper | ACTGGTACCTTCA CGCAGCGTGGTCAGTC | Used with EcoRI site in TOPO to construct sapM-lacZ reporter pBM94 |
| Recombinant DNA reagent (primers) | PsapM-LacZ R | this paper | AGGATCCATTCCGC GGAGCATGCCGGGAG | Contains engineered BamHI site to construct sapM-lacZ reporter pBM94 |
| Recombinant DNA reagent (primers) | SapM-3311 gap F | this paper | GACGGGTTAT GCGACCAATG | Amplify the region between sapM and satS for RT-PCR |
| Recombinant DNA reagent (primers) | SapM-3311 gap R | this paper | CTCAAGCGGA TGGGTACGAG | Amplify the region between sapM and satS for RT-PCR |

*Continued on next page*

*Continued*

| Reagent type (species) or resource | Designation | Source or reference | Identifiers | Additional information |
|---|---|---|---|---|
| Recombinant DNA reagent (primers) | Mce4A hsp60 F | this paper | AAGATATCCGAA CGGAAACGCCAAACG | Contains engineered EcoRV site, used to construct mce4AMsmeg-HA plasmids pBM44 |
| Recombinant DNA reagent (primers) | Mce4A hsp60 R | this paper | TAAGCTTCGTCC CTTTCCGCGAAC | Contains engineered HindIII site, used to construct mce4A Msmeg-HA plasmids pBM44 |
| Recombinant DNA reagent (primers) | sigA RT F | this paper | AAGCGAACAG CGGCGAAGTC | qRT-PCR primer for sigA |
| Recombinant DNA reagent (primers) | sigA RT R | this paper | TTCGGGATGG TGCTGGTCGTAG | qRT-PCR primer for sigA |
| Recombinant DNA reagent (primers) | sapM RT F | this paper | ATCGTTGCTGG CCTCATGG | qRT-PCR primer for sapM |
| Recombinant DNA reagent (primers) | sapM RT R | this paper | AGGGAGCCGA CTTGTTACC | qRT-PCR primer for sapM |
| Recombinant DNA reagent (primers) | sapM E. coli F | this paper | GTCTCTCCCATGC TCCGCGGAATCCAG | Used to express sapM in the E. coli pMSCG -28 vector |
| Recombinant DNA reagent (primers) | sapM E. coli R | this paper | GGTTCTCCCCAG CGTCGCCCCAAATAT CGGTTATTGG | Used to express sapM in the E. coli pMSCG-28 vector |
| Recombinant DNA reagent (primers) | satS E. coli F | this paper | TTTTTTCATATGGTT GCTGACCT CGTACCCATC | Contains engineered NdeI site, used to express satS in E. coli Pet28b vector |
| Recombinant DNA reagent (primers) | satSC E. coli F | this paper | TTTTTTCATATGCG GGACTTCTGGTTGCAG | Contains engineered NdeI site, used to express satSC in E. coli Pet28b vector |
| Recombinant DNA reagent (primers) | satS/satSC E. coli R | this paper | TTTTTTAAGCTTCT ATTCGCGCCTGAGCC | Contains engineered HindIII site, used to express satS/satSC in E. coli Pet28b vector |
| Recombinant DNA reagent (plasmids) | pMV261.kan | *Stover et al., 1991* | | Multicopy mycobacterial vector with hsp60 promoter (KanR) |
| Recombinant DNA reagent (plasmids) | pMV361.kan | *Stover et al., 1991* | | Single-copy mycobacterial vector with hsp60 promoter, integrates in mycobacteriophage L5 attB site (KanR) |

Continued

| Reagent type (species) or resource | Designation | Source or reference | Identifiers | Additional information |
|---|---|---|---|---|
| Recombinant DNA reagent (plasmids) | pMV306.kan | *Stover et al., 1991* | | Single-copy, promoterless mycobacterial vector, integrates in mycobacteriophage L5 attB site (KanR) |
| Recombinant DNA reagent (plasmids) | pJSC77 | *Glickman et al., 2000* | | Multicopy mycobacterial vector, HA tag cloned into pMV261 (KanR) |
| Recombinant DNA reagent (plasmids) | pLL2 | *Ligon et al., 2013* | | single-copy mycobacterial shuttle vector, integrates in mycobacteriophage Tweety attB site (HygR) |
| Recombinant DNA reagent (plasmids) | pYA810 | *Gibbons et al., 2007* | | Integrating M. smegmatis secA2 complementation plasmid in pMV361.kan (KanR) |
| Recombinant DNA reagent (plasmids) | pNR25 | *Rigel et al., 2009* | | Integrating M. smegmatis secA2 K129R in pMV361.kan (KanR) |
| Recombinant DNA reagent (plasmids) | pLL50 | this paper | | Suicide vector pMP62 containing flanking regions to delete satSMsm (HygR) |
| Recombinant DNA reagent (plasmids) | pBM11 | this paper | | Suicide vector pMP62 containing secA2Msm and flanking regions to reintroduce secA2 to the BAF1 strain (HygR) |
| Recombinant DNA reagent (plasmids) | pBM4 | this paper | | satSMsm under native promoter in pLL2 (HygR) |
| Recombinant DNA reagent (plasmids) | pBM80 | this paper | | satSMtb under hsp60 promoter in pLL2 (HygR) |
| Recombinant DNA reagent (plasmids) | pSM42 | this paper | | satSMtb upstream and downstream flanks inserted into pYUB854 (HygR) |
| Recombinant DNA reagent (plasmids) | pSM45 | this paper | | Phasmid for knocking out satSMtb (HygR) |
| Recombinant DNA reagent (plasmids) | pSM60 | this paper | | Phage for knocking out satSMtb (HygR) |
| Recombinant DNA reagent (plasmids) | pBM13 | this paper | | satSMtb under hsp60 promoter in pMV306.kan (KanR) |
| Recombinant DNA reagent (plasmids) | pJTS130 | *Zulauf et al., 2018* | | sapM under hsp60 promoter in pMV261.kan (KanR) |

*Continued*

| Reagent type (species) or resource | Designation | Source or reference | Identifiers | Additional information |
|---|---|---|---|---|
| Recombinant DNA reagent (plasmids) | pJTS132 | this paper | | Δss-sapM under hsp60 promoter in pMV261. kan (KanR) |
| Recombinant DNA reagent (plasmids) | pYUB76 | *Barletta et al., 1992* | | Multicopy mycobacterial shuttle vector with promoterless lacZ gene (KanR) |
| Recombinant DNA reagent (plasmids) | pBM94 | this paper | | psapM-sapM'-'lacZ in pYUB76 (KanR) |
| Recombinant DNA reagent (plasmids) | pBM56 | this paper | | sapM under hsp60 promoter in pMV261.kan containing a C-terminal linker and FLAG tag (KanR) |
| Recombinant DNA reagent (plasmids) | pBM60 | this paper | | satSMtb under hsp60 promoter in pLL2 containing a C-terminal HA tag (HygR) |
| Recombinant DNA reagent (plasmids) | pBM61 | this paper | | Δss-sapM under hsp60 promoter in pMV261.kan containing a C-terminal linker and FLAG tag (KanR) |
| Recombinant DNA reagent (plasmids) | pBM22 | this paper | | satSMsm under native promoter in pLL2 amplified from suppressor 3S to contain the G134D point mutation (HygR) |
| Recombinant DNA reagent (plasmids) | pBM23 | this paper | | satSMsm under native promoter in pLL2 amplified from suppressor 3S to contain the G134D point mutation and containing a C-terminal HA tag (HygR) |
| Recombinant DNA reagent (plasmids) | pBM87 | this paper | | satSMtb under hsp60 promoter in pLL2 with point mutation G134D generated by site directed mutagenesis (HygR) |
| Recombinant DNA reagent (plasmids) | pBM44 | this paper | | Mce4AMsm under hsp60 promoter in pJSC77 containing a C-terminal HA tag (KanR) |
| Recombinant DNA reagent (plasmids) | pHSG58 | *Gibbons et al., 2007* | | Multi-copy Ms1704-HA expression vector under hsp60 promoter (KanR) |
| Recombinant DNA reagent (plasmids) | pRH1 | this paper | | sapM in the E. coli pMSCG-28 vector containing a C terminal His tag (CarbR) |

*Continued on next page*

*Continued*

| Reagent type (species) or resource | Designation | Source or reference | Identifiers | Additional information |
|---|---|---|---|---|
| Recombinant DNA reagent (plasmids) | pRH2 | this paper | | satS in E. coli Pet28b vector (KanR) |
| Recombinant DNA reagent (plasmids) | pRH3 | this paper | | satSC in E. coli Pet28b vector (KanR) |
| Antibody | Rabbit polyclonal anti-SatS | this paper | rabbit polyclonal raised against SatS*Mtb*: PA6753 for *Mtb* and PA6754 for *Msm* | (1:20,000) |
| Antibody | Rabbit polyclonal anti-SapM | *Vergne et al., 2005* | | Provided by Vojo Deretic; (1:5,000) |
| Antibody | Rabbit polyclonal anti-Mce1A | *Feltcher et al., 2015* | | Provided by Chris Sassetti; (1:10,000) |
| Antibody | Rabbit polyclonal anti-Mce1E | *Feltcher et al., 2015* | | Provided by Chris Sassetti; (1:10,000) |
| Antibody | Rabbit polyclonal anti-Mce1D | *Perkowski et al., 2016* | | Provided by Chris Sassetti; (1:10,000) |
| Antibody | Rabbit polyclonal anti-19kDa | *Feltcher et al., 2015* | | Provided by Douglas Young; (1:20,000) |
| Antibody | Mouse monoclonal anti-PhoS1 | NIH Biodefense and Emerging Infections Research Resources Repository, NIAID | Cat. #: IT23 | (1:20,000) |
| Antibody | Rabbit polyclonal anti-PknG | *Feltcher et al., 2015* | | Provided by Yossef Av-Gay; (1:5,000) |
| Antibody | Rabbit polyclonal anti-SecA2 | *Rigel et al., 2009* | | (1:20,000) |
| Antibody | Rabbit polyclonal anti-SigA | *Feltcher et al., 2015* | | Provided by Murty Madiraju; (1:15,000) |
| Antibody | Rabbit polyclonal anti-MspA | *Feltcher et al., 2013* | | Provided by Michael Niederweis; (1:5,000) |
| Antibody | Rabbit polyclonal anti-SecY | *Ligon et al., 2013* | | (1:250) |
| Antibody | Rabbit polyclonal anti-Mpt32 | NIH Biodefense and Emerging Infections Research Resources Repository, NIAID | Cat. #: NR-13807 | (1:5,000) |
| Antibody | Rabbit polyclonal anti-FLAG | Sigma-Aldrich | Cat. #: F7425 | (1:10,000) |
| Antibody | Mouse monoclonal anti-FLAG | Sigma-Aldrich | clone M2 | (1:10,000) |
| Antibody | Mouse monoclonal anti-HA | Sigma-Aldrich | clone HA-7 | (1:10,000) |
| Antibody | Mouse monoclonal anti-HIS | Abgent | Cat. #: AM1010a | (1:20,000) |
| Antibody | Goat polyclonal anti-Mouse IgG | Bio-Rad | Cat. #: 1721011 | (1:25,000) |

*Continued on next page*

*Continued*

| Reagent type (species) or resource | Designation | Source or reference | Identifiers | Additional information |
|---|---|---|---|---|
| Antibody | Goat polyclonal anti-Rabbit IgG | Bio-Rad | Cat. #: 1706515 | (1:25,000) |
| Chemical compound, drug | p-Nitrophenyl Phosphate (PNPP) | NEB | Cat. #: P0757 | |
| Software, algorithm | ImageJ | https://imagej.nih.gov/ij/ | RRID:SCR_003070 | |
| Software, algorithm | Graphpad Prism 7 | https://www.graphpad.com/scientific-software/prism/ | RRID:SCR_002798 | |

## Plasmids, bacterial strains, and culture conditions

For plasmid construction, PCR products were amplified with primers described in the Key Resources Table, ligated into TOPO cloning vectors (Invitrogen, Carlsbad, CA), digested with restriction enzymes, and ligated into their final vectors. Final vectors are described in the Key Resources Table. In all cases, newly constructed plasmids were verified by sequencing and diagnostic digests. In the case of SatS G134D plasmids, $satS_{Msm}$ was amplified by PCR from the 3S suppressor and $satS_{Mtb}$ G134D was designed using site directed mutagenesis (SDM) on the $satS_{Mtb}$ complementation plasmid pBM81. Amino acid G134 was confirmed to be highly conserved in mycobacterial SatS homologs using ConSurf (*Ashkenazy et al., 2016*).

*M. tuberculosis* and *M. smegmatis* strains are described in the Key Resources Table. For all experiments in this study, wildtype and mutant strains had empty vector plasmids to enable comparison to complemented strains. *M. tuberculosis* was grown at 37°C in Middlebrook 7H9/7H11 supplemented with 1x albumin dextrose saline (ADS), 0.5% glycerol and 0.025% Tween 80. *M. smegmatis* was grown at 37°C or 30°C in Middlebrook 7H9/7H10 or Mueller-Hinton medium. Media were supplemented with 0.5% glycerol plus 0.2% glucose (7H9/7H10 medium only) and 0.05% Tween 80 (all media). For all mycobacteria, the antibiotics kanamycin (20 μg/mL) and hygromycin B (50 μg/mL) were added as needed. *E. coli* strains were grown at 37°C in Miller LB broth or on Miller LB agar. The antibiotics kanamycin (40 μg/mL) and hygromycin B (150 μg/mL) were added as needed.

*M. smegmatis* growth was monitored using resazurin. At an $OD_{600\ nm}$ of 1, cells were diluted to $10^5$ c.f.u. $ml^{-1}$ in the same medium and 100 μl were added to 96-well plates. After 24 hr of growth at 37°C, resazurin (12.5 μg ml−one final concentration; Sigma-Aldrich, St. Louis, MO) was added and fluorescence with excitation at 530 nm and emission at 590 nm was monitored over time. *M. tuberculosis* growth was monitored by measuring the optical density (OD600) of liquid broth cultures over time.

## Suppressor collection and sequencing

The suppressor screen was performed as described previously (*Ligon et al., 2013*). Suppressors of the *secA2 K129R* allele were isolated by plating independently grown cultures of the *secA2 K129R* strain onto Mueller-Hinton agar at 37°C. Genomic DNA from six suppressors was submitted for whole genome sequencing at the High-Throughput Sequencing Facility at the University of North Carolina at Chapel Hill. Sequencing was performed using Illumina GA II technology. Reads were aligned to the *M. smegmatis* mc²155 reference genome (NCBI RefSeq accession number NC_008596.1) using SOAP (*Li et al., 2008*).

## *M. smegmatis* mutant construction

The *M. smegmatis* unmarked Δ*secA2*/Δ*satS* double mutant was created by two-step allelic exchange using plasmid pLL50 in the Δ*secA2* mutant strain NR116, resulting in strain BAF1. Briefly, the suicide plasmid pLL50, containing a hygromycin-resistance selectable marker, a *sacB* counter-selectable marker, and flanking regions for *satS* was transformed into *M. smegmatis*. Transformants were selected by plating on media containing hygromycin B. Hygromycin-resistant transformants were

grown to saturation, diluted 1:100 in media lacking hygromycin B, and then grown overnight at 37°C. Bacteria in which a second recombination event occurred were selected by plating on 7H10 supplemented with 0.2% glucose and 4.5% sucrose. BAF1 was assessed for the desired chromosomal deletion by PCR and Southern blot.

The *M. smegmatis* unmarked Δ*satS* single mutant was created by adding back *secA2* into the BAF1 strain by two-step allelic exchange using plasmid pBM11, resulting in strain BM10. BM10 was assessed for the desired chromosomal insertion by PCR and Southern blot. Additionally, immunoblots of SecA2 were performed to ensure SecA2 levels were fully restored.

## *M. tuberculosis* mutant construction

The *satS* deletion mutant was created in H37Rv using the specialized transducing phage system as previously described (*Braunstein et al., 2001*). Briefly, cosmid pSM42 was created by subcloning *satS* upstream and downstream flanks into pYUB854 surrounding the hygromycin cassette. Cosmid pSM42 was ligated into phAE159 to generate recombinant phasmid pSM45. The recombinant phasmid, pSM45 was packaged into phage head using a λ in vitro packaging extract kit (Gigapack III XL, Agilent, Santa Clara, CA) and was transduced into *E. coli*. Phasmid DNA was electroporated into *M. smegmatis* mc²155 to make phage (pSM60). Transduced phage was plaque purified and amplified for high titer phage lysate. H37Rv was transduced with high phage lysate as previously described (*Braunstein et al., 2001*). Transductants were grown at 37°C on Middlebrook 7H10 plates containing hygromycin for 4 weeks. To confirm the *satS* deletion in transductants, PCR and Southern blotting were used.

## Azide sensitivity assays

Cultures were plated for azide sensitivity as previously described, by mixing 200 μL of a saturated culture with 7H9 top agar and pouring over 7H10 agar plates lacking tween in three technical replicates (*Ligon et al., 2013*). The diameter of the zone of inhibition was measured after two days and reported as a percentage of the entire plate diameter, yielding percent azide inhibition.

## Subcellular fractionation and immunoblotting

When whole cell lysates were prepared from the same cultures used to isolate culture filtrate proteins, exponential phase *M. tuberculosis* cultures grown in Sauton media without detergent were fixed in an equal volume of 10% formalin for 1 hr. Fixed cells were pelleted by centrifugation, resuspended in extraction buffer, and lysed by bead beating.

When prepared for subcellular fractionation, cultures of *M. smegmatis* and irradiated *M. tuberculosis* grown in Middlebrook 7H9 medium were isolated as previously described (*Perkowski et al., 2016*; *Feltcher et al., 2013*). Briefly, cells suspended in 1X PBS containing protease inhibitors were lysed by passage through a French pressure cell. Unlysed cells were removed by centrifugation at 3000 x *g* for 30 min to generate clarified whole cell lysates (WCLs). The WCLs were either spun at 100,000 x *g* for 2 hr to collect the cell envelope fraction containing both the cell wall and membrane (ENV) or at 27,000 x *g* for 30 min to pellet the cell wall fraction only (CW). The supernatant following CW isolation was spun at 100,000 x *g* for 2 hr to separate the membrane fraction (MEM) and collect the soluble cytoplasm-containing fraction (SOL). Protein concentrations were determined by bicinchoninic acid assay (Pierce, ThermoFisher, Waltham, MA).

Samples containing equal protein were separated by SDS-PAGE and transferred to nitrocellulose membranes. After blocking, proteins were detected using antibodies described in the Key Resources Table.αHis (Abgent, San Diego, CA) was used to detect the mycobacterial GroEL1 which has a string of endogenous histidines. αMouse and αRabbit IgG conjugated horseradish peroxidase secondary antibodies (Bio-Rad, Hercules, CA) were used and signal was detected using Western Lightning Plus-ECL chemiluminescent detection reagent (Perkin-Elmer, Waltham, MA).

## Culture filtrate protein preparation

M.*M. tuberculosis* culture filtrates were collected as described previously (*Zulauf et al., 2018*). Briefly, 200 mL cultures were grown in Sauton media at 37°C for 24 hr. The supernatants were double filtered through a 0.2 μm filter. Supernatant proteins were concentrated 200 fold using 3,000 MW cut off centrifuge filters (Amicon) by centrifugation at 3,000 rpm at 4°C. For immunoblotting,

protein was precipitated overnight at 4°C with 10% trichloroethanoic acid (TCA). Protein pellets were washed with acetone, resuspended in 250 uL of 1 x SDS-PAGE buffer.

For *M. smegmatis* culture filtrate collection, samples were obtained as previously described (*Feltcher et al., 2013*). In brief, 10 mL cultures were grown without Tween 80 to an $OD_{600\ nm}$ of 0.4 to 0.7. Supernatant was separated from cells first by centrifugation at 3000 x g and then filtration through a 0.2-µm-pore-size filter. Protein from 2 mL of supernatant was TCA precipitated as described above and then resuspended in 50 µL of $1 \times$ SDS PAGE buffer.

## Phosphatase activity assay

SapM activity was assayed as described previously (*Saleh and Belisle, 2000*; *Zulauf et al., 2018*). In a 96 well plate, 3 µg of CFP protein was diluted with water and added to 10X buffer (1M Tris base pH 6.8) with 20 mM Sodium tartrate to reduce background phosphatase activity, and 50 mM p-nitro-phenyl phosphate (pNPP) for a total volume of 200 µL (New England Biolabs, Ipswich, MA). Tartrate is an inhibitor of some phosphatases, but SapM activity is unaffected by tartrate (*Saleh and Belisle, 2000*). Despite the addition of tartrate, the phosphatase assay used is not specific for SapM. The residual activity in the Δ*secA2* and Δ*satS* mutants can be attributed to SatS-independent phospha-tases. The plate was incubated at 37°C in a plate reader, and the absorbance at 405 nm was mea-sured every three minutes for four hours. Over the linear portion of the kinetic assay, we calculated the rate of pNPP conversion by calculating the slope of the line generated by plotting $Abs_{405\ nm}$ as a function of time. These slopes were then normalized to the WT rate of change, which we set to 100%.

## Whole cell phosphatase activity assay

To perform the whole cell phosphatase activity assay, *M. smegmatis* strains expressing SapM (±ss) or an empty vector were grown in 7H9 medium to an $OD_{600}$ of 1, pelleted, and washed once in 7H9 medium. Cells were diluted to $6.25 \times 10^5$ CFU/mL in 7H9 medium and 160 µL was added in tripli-cate to a 96-well plate. Plates were incubated at 37°C for 24 hr. After 24 hr, 20 µL of 10X buffer (1M Tris base pH 6.8) with 20 mM Sodium tartrate and 50 mM p-nitrophenyl phosphate (pNPP) were added to the wells for a total volume of 200 µL. The plate was incubated at 37°C in a plate reader, and the absorbance at 405 nm was measured every three minutes for four hours. We calculated the rate of pNPP conversion as described above.

## Macrophage infection

To assess *M. tuberculosis* survival in macrophages, $2 \times 10^5$ BMDMs from C57BL/6 mice were seeded 1 day prior to infection with *M. tuberculosis* (H37Rv, Δ*secA2*, Δ*satS*, or Δ*satS +psatS*) at an MOI of 1 as previously described (*Sullivan et al., 2012*; *Zulauf et al., 2018*). At 4 hr post infection, macro-phages were washed four times and at the indicated time points were lysed with 0.1% Triton X-100. Serial dilutions of the lysates were plated on 7H11 agar plates and CFUs were counted three weeks later.

## Reverse transcriptase-PCR

To assess the operon nature of *sapM* and *satS*, RNA was extracted from mid-log phase cultures of *M. tuberculosis* H37Rv (see *qRT-PCR* for Materials and methods). Reverse transcription reaction was carried out using iScript cDNA Synthesis Kit (Bio-Rad) and random primers. PCR amplification of the intergenic regions on cDNA were performed using specific primers on *sapM* and *satS* (Key Resour-ces Table). Controls included primers for the housekeeping gene *sigA*, PCR amplification from geno-mic DNA, and PCR amplification from RNA lacking reverse transcriptase.

## Quantitative Real-Time PCR

Triplicate *M. tuberculosis* cultures were grown in modified 7H9 medium to an $OD_{600}$ of 1 and RNA was isolated as previously described using a chloroform-methanol and Trizol (Invitrogen) extraction (*Perkowski et al., 2016*; *Feltcher et al., 2015*). RNA samples were treated with DNase (Promega, Madison, WI) and then column purified (Zymo RNA clean and concentrator Kit, Irvine, CA). Following RNA isolation, cDNA was synthesized with random primers using the iScript cDNA Synthesis Kit (Bio-Rad). Real-time PCR was completed using 25 ng of cDNA template in triplicate technical

replicates using the SensiMix SYBR and fluorescein kit (Bioline, Toronto, Canada). Transcripts were normalized to the housekeeping gene *sigA*. Primer sequences are provided in the Key Resources Table.

## LacZ (β-galactosidase) activity assays

LacZ activity assays in *M. tuberculosis* were performed using a modified protocol previously described for *M. smegmatis* (*Ligon et al., 2013*). Strains were grown in 7AGT to mid-log phase and 800 μL was pelleted. Pellets were resuspend in 800 μL Z buffer (60 mM $Na_2HPO_4$, 40 mM $NaH_2PO_4$, 10 mM KCl, 1 mM $MgSO_4$, 50 mM β-mercaptoethanol), then lysed with 35 μL chloroform and 1 μL of 0.1% SDS by vortexing for 30 s followed by sonication. 640 μg of *o*-nitrophenyl-β-D-galatopyranoside was added to each reaction and mixtures were incubated for 24 min at room temperature. Reactions were terminated by addition of 400 μL of 1 M $Na_2CO_3$. Debris was removed by centrifugation at 3,000 rpm for 10 min, and the $OD_{420 nm}$ was read from the supernatant. LacZ activity (Miller units) was calculated by the following formula: $(1000 \times OD_{420 nm})/([$reaction time in minutes$] \times [$culture volume used in the reaction, in mL$] \times OD_{600 nm})$.

## SatS antiserum production

To generate polyclonal antisera against SatS, purified $SatS_{Mtb}$ was produced in *E. coli* and injected into two rabbits using Titermax adjuvant (ThermoFisher). The serum from both rabbits was tested against wild-type *M. tuberculosis* and *M. smegmatis* and the Δ*satS* mutants for specificity. The serum from rabbit PA6753 recognizes $SatS_{Mtb}$ but does not recognize $SatS_{Msm}$ and is only used for $SatS_{Mtb}$. The sera from rabbit PA6754 has a non-specific band at the same size as $SatS_{Mtb}$, but recognizes $SatS_{Msm}$ and is only used for $SatS_{Msm}$.

## Co-immunoprecipitation

For in vivo co-immunoprecipitation, *M. smegmatis* cells were transformed with $SatS_{Mtb}$ (±HA) tag and SapM-FLAG (±signal sequence). Transformed cells were grown in 50 mL of 7H9 medium to an $OD_{600 nm}$ of 0.5. Cells were pelleted and resuspended in 2.5 mL 1X PBS buffer containing a protease inhibitor cocktail. Cells were lysed by passage through a French pressure cell. Unlysed cells were removed by centrifugation at 3000 x *g* for 30 min to generate clarified whole cell lysates (WCLs). 200 μL of lysate was diluted in 1 mL of 1X PBS + protease inhibitors, added to 25 μL anti-HA agarose (Sigma-Aldrich), and mixed end to end at 4°C for 4 hr, followed by four washes with 1X PBS. The immunoprecipitated SatS-HA along with co-immunoprecipitated proteins were eluted in 25 μL of 1X SDS-PAGE buffer, run on 15% SDS-PAGE gels for 4.5 hr, transferred onto nitrocellulose membranes, and immunoblotted.

## Cloning, expression, and purification of SapM inclusion bodies (IBs)

The *sapM* full length gene was PCR amplified from genomic DNA of H37Rv using Phusion high-fidelity DNA polymerase and the primers sapM *E. coli* F and sapM *E. coli* R. The resulting PCR product treated with T4 polymerase and mixed with linear, T4 treated, pMSCG-28 vector, and transformed into chemically competent BL21 (DE3) cells as previously described (*Eschenfeldt et al., 2010*).

SapM containing a C-His$_6$-tag and tobacco etch virus (TEV) protease cleavage site were grown in Luria-Bertani (LB) broth containing 100 μg/mL carbenicillin at 37°C to an OD of 0.8 ($A_{600 nm}$). SapM expression was induced with the addition of 0.5 mM isopropyl-β-D-thiogalactoside (IPTG) and cells were grown for an additional 5 hr at 37°C. Cells were harvested and resuspended in lysis buffer (40 mM HEPES pH 7.4, 300 mM NaCl, 10 mM imidazole). Cells were broken using a high-pressure homogenizer in the presence of protease inhibitor cocktail (EMD Millipore, Burlington, MA) and centrifuged at 30,000 x *g*.

In order to clarify SapM IBs, a modified protocol was adopted (*Palmer and Wingfield, 2004*). The supernatant obtained after cell lysis was decanted and the pellet resuspended in 40 mM HEPES pH 7.4, 2% Triton X-100, 5 mM EDTA. The suspension was then homogenized using sonication for 3 cycles for 30 s each, centrifuged at 30,000 x *g* for 15 min, and the supernatant decanted. This was repeated three times to remove cell wall, membrane material, and lipid/membrane associated proteins. In the final step, detergent was omitted, and SapM purity of greater than 95% was confirmed via SDS-PAGE.

## Protein aggregation assay

Inclusion bodies of SapM pre protein with a 6X C-terminal His tag were denatured in 8 M urea, 40 mM HEPES pH 7.4, 100 mM NaCl, 1 mM EDTA to a final concentration of 150 µM. Denatured SapM (1 µL) was rapidly diluted into buffer (150 µL) containing 40 mM HEPES pH 7.4, 100 mM NaCl, and 1 mM EDTA. Protein aggregation was monitored in the absence or presence of SatS at 25°C by measuring light scattering in a time dependent manner using a Cary Eclipse Varian with excitation and emission at 350 nm.

## Cloning, expression, purification, and crystallization of SatS and $SatS_C$

The $satS$ and $satS_C$ genes were PCR amplified from genomic DNA of H37Rv using Phusion high-fidelity DNA polymerase and the primers $satS$ E. coli F, $satS_C$ E. coli F, and $satS/satS_C$ E. coli R. The resulting PCR products were digested with NdeI and HindIII, ligated into Nde/HindIII digested Pet28b vector, and transformed into chemically competent BL21 (DE3) cells.

SatS and $SatS_C$ with a N-His$_6$-tag and TEV protease cleavage site were grown separately in LB broth containing 50 µg/mL kanamycin at 37°C to an OD of 0.8 ($A_{600\ nm}$). SatS and $SatS_C$ expression was induced with the addition of 0.5 mM IPTG and cells were grown for an additional 5 hr at 37°C. Cells were harvested and resuspended in lysis buffer (40 mM HEPES pH 7.4, 300 mM NaCl, 10 mM imidazole). Cells were broken using a high-pressure homogenizer in the presence of protease inhibitor cocktail (EMD Millipore) and centrifuged at 30,000 x $g$. SatS and $SatS_C$ were purified using a cOmplete His-tag purification resin (Roche, Basel, Switzerland), followed by removal of the tag using TEV protease at 25°C and further purified by size exclusion chromatography using a Superdex 200 26/60 (GE Healthcare, Chicago, IL). Protein purity was greater than 95% as determined by SDS-PAGE. Protein concentration was measured spectrophotometrically at 595 nm using Bradford reagent.

Crystals of SatS (20 mg/mL) were produced after screening 768 individual conditions using sitting drop vapor diffusion method at 16°C with a 50 µL well solution and a drop consisting of 1.2 µL of 0.6 µL protein and 0.6 µL of well solution. A single diffraction quality crystal appeared within 6 months in 3.5 M ammonium sulfate and 0.1 M sodium acetate trihydrate pH 4.6. SatS was indexed into space group $P2_12_12_1$ with the unit cell parameters a = 50, b = 51, c = 76. The unit cell was comprised of a single molecule in the asymmetric unit.

Crystals of $SatS_C$ (12 mg/mL) were produced after screening 384 individual conditions using sitting drop vapor diffusion method at 16°C with a 50 µL well solution and a drop consisting of 1.2 µL of 0.6 µL protein and 0.6 µL of well solution. Initial crystal hits were optimized using hanging drop vapor diffusion method at 16°C with a 1 mL well solution and a 4.0 µL drop consisting of random ratios of protein to well solution. The highest quality crystals appeared overnight in 3.5 M ammonium citrate pH 6.4 and continued to mature for an additional 2 weeks. $SatS_C$ indexed into space group $P2_12_12_1$ with the unit cell parameters a = 50, b = 51, c = 76. The unit cell was comprised of a single molecule in the asymmetric unit.

## Data collection and structure determination of $SatS_C$

X-ray diffraction data were collected from a single crystal at beamline 23-ID of the GM/CA-CAT facilities of the Advanced Photon Source, Argonne National Laboratory. The structure of SatS was solved by single-wavelength anomalous dispersion (SAD) using a Bromine (Br) derivative. The data were processed and reduced using the HKL3000 software package. A single Br site was identified using Phenix HySS, and Phenix AutoSol was used to produce the initial electron density map. Simultaneous rounds of model building and structure refinement were performed manually in Coot and Phenix Refine. Additional structures of $SatS_C$ were solved by molecular replacement using the initial structure of $SatS_C$ as a model in Phenix Phaser MR. Simultaneous rounds of model building and structure refinement were carried out in Coot and Phenix Refine.

## Statistical analyses

For comparisons between the groups for the determination of (i) phosphatase activity in the mycobacterial culture filtrates, (ii) whole cell phosphatase activity (iii) '$lacZ$ reporter fusions and (iv) growth in macrophages, one-way analysis of variance (ANOVA) with the Tukey post test was employed. For

the statistical analysis and generation of graphs, Prism five software (version 7; GraphPad Software Inc., CA) was used.

## Acknowledgements

We thank Carol Teshke for critical reading of this manuscript and providing us with the purified SatS protein used to generate antibodies. We also thank the Braunstein laboratory and Martin Pavelka for experimental advice and critical reading of this manuscript as well as Vojo Deretic, Yossef Av-Gay, Douglas Young, Murty Madiraju, Chris Sassetti, and Michael Niederweis for kindly providing us with antibodies. This work was supported by National Institutes of Health/National Institute of Allergy and Infectious Diseases grant R01 AI054540 and grant P01 AI095208 as well as Welch Foundation grant A-0015. BK.M was supported by a University of North Carolina Dissertation Completion Fellowship. BRAJ was supported by an Initiative for Maximizing Student Diversity (IMSD) grant from NIGMS (R25-GM055336).

## Additional information

### Funding

| Funder | Grant reference number | Author |
|---|---|---|
| National Institute of Allergy and Infectious Diseases | AI054540 | Brittany K Miller<br>Lauren S Ligon<br>Nathan W Rigel<br>Seidu Malik<br>Miriam Braunstein |
| University of North Carolina | Graduate School Disseration Award | Brittany K Miller |
| National Institute of Allergy and Infectious Diseases | R21 AI135899 | Brittany K Miller<br>Miriam Braunstein |
| National Institute of Allergy and Infectious Diseases | P01 AI095208 | Ryan Hughes |
| Welch Foundation | A-0015 | Ryan Hughes<br>James C Sacchettini |
| National Institute of General Medical Sciences | GM055336 | Brandon R Anjuwon-Foster |

The funders had no role in study design, data collection and interpretation, or the decision to submit the work for publication.

### Author contributions

Brittany K Miller, Conceptualization, Data curation, Investigation, Methodology, Writing—original draft, Writing—review and editing; Ryan Hughes, Data curation, Investigation, Methodology, Writing—review and editing; Lauren S Ligon, Data curation, Investigation, Writing—review and editing; Nathan W Rigel, Seidu Malik, Brandon R Anjuwon-Foster, Data curation, Writing—review and editing; James C Sacchettini, Supervision, Funding acquisition, Writing—review and editing; Miriam Braunstein, Conceptualization, Supervision, Funding acquisition, Methodology, Writing—original draft, Writing—review and editing

### Author ORCIDs

Brittany K Miller (iD) http://orcid.org/0000-0003-2093-4436
Miriam Braunstein (iD) http://orcid.org/0000-0003-1180-0030

### Ethics

Animal experimentation: All animal care and experimental protocols were in strict accordance with the NIH Guide for the Care and Use of Laboratory Animals and were approved by the Institutional Animal Care and Use Committee of the University of North Carolina (protocol number 15-018.0).

Decision letter and Author response
Decision letter https://doi.org/10.7554/eLife.40063.027
Author response https://doi.org/10.7554/eLife.40063.028

## Additional files

### Supplementary files
• Transparent reporting form
DOI: https://doi.org/10.7554/eLife.40063.020

### Data availability
All data generated and analysed during this study are included in the manuscript and supporting files. Figure supplements have been provided for Figures 1, 2, 5, and 8. Two additional supplementary tables describe the primers and plasmids used in this study. SatS C domain X-ray structure validation details are described in Figure 8—figure supplement 1 and have been deposited in PDB under the accession codes 6DRQ and 6DNM.

The following datasets were generated:

| Author(s) | Year | Dataset title | Dataset URL | Database and Identifier |
|---|---|---|---|---|
| Hughes RC, Sacchettini JC | 2019 | The crystal structure of SatS c-terminal domain in complex with bromine | https://www.rcsb.org/structure/6DRQ | Protein Data Bank, 6DRQ |
| Hughes RC | 2019 | The crystal structure of SatS c-terminal domain | https://www.rcsb.org/structure/6DNM | Protein Data Bank, 6DNM |

The following previously published dataset was used:

| Author(s) | Year | Dataset title | Dataset URL | Database and Identifier |
|---|---|---|---|---|
| Dekker C, de Kruijff B, Gros P | 2003 | Crystal Structure of SecB from Escherichia coli | https://www.rcsb.org/structure/1QYN | Protein Data Bank, 1QYN |

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
