## [Decision Letter]

Thank you for submitting your article "*Mycobacterium tuberculosis* SatS is a chaperone for the SecA2 protein export pathway" for consideration by *eLife*. Your article has been reviewed by three peer reviewers, one of whom is a member of our Board of Reviewing Editors, and the evaluation has been overseen by Gisela Storz as the Senior Editor. The reviewers have opted to remain anonymous.

The reviewers have discussed the reviews with one another and the Reviewing Editor has drafted this decision to help you prepare a revised submission.

Summary:

The submission by Braunstein and colleagues describes an investigation of secretion through the specialized SecA2 translocation system in mycobacteria. Using a mutant with a K129R mutation in SecA2, the authors isolate suppressors of the growth defect in this strain that mapped to Msmeg_1684, with mutations in various regions of this gene. The authors annotate this gene as SatS and confirm that deletion thereof relieved defects of the *secA2 K129R* mutant. Using heterologous complementation experiments, the authors also confirm that the SatS proteins from *M. tuberculosis* and *M. smegmatis* are functionally interchangeable. Loss of SatS was able to correct mis-localization of the SecA2 K129R mutant protein, which was found predominantly in the soluble fraction in the SatS deletion mutant. Deletion of *satS* alleviated the SecY abundance defects in the SecA2 K129R mutant. Next, the authors delete *satS* in *M. tuberculosis* and report a dramatic reduction in SapM secretion and, as SapM has phosphatase activity, reduced phosphatase activity in the culture filtrate of the mutant. Similar results were noted when the *M. tuberculosis* SapM was expressed in *M. smegmatis.* Following this, the authors confirm that Mce1, Mce1E and Mce4A-HA (but not PknG, PhoS1 or Msmeg1704-HA) required SatS for transport. The authors note that deletion of SatS resulted in reduced cytoplasmic levels of some proposed targets, suggesting that SatS post-translationally stabalizes its targets, possibly serving as an export chaperone. This hypothesis was supported by the demonstration that SatS co-precipitates with one of its targets, SapM; this interaction was not dependent on the signal sequence of SapM. Using a *satS secA2* double mutant, the authors confirm that SatS most likely stabilizes substrates before they are transported by the SecA2 translocon. With recombinant forms of SatS and SapM, the authors confirm that SatS prevented aggregation of SapM, much like a chaperone acting on its natural substrate. Structural analysis of the SatS C-terminal domain suggested a novel fold, with an electronegative charge, similar to other reported chaperones. Consistent with this, the C-terminal domain was sufficient to prevent aggregation of recombinant SapM. The authors further interrogate the role of the G143 residue in SatS and their observations suggest that SatS has more than one role in secretion of a select set of SecA2 substrates.

Key findings:

1) Deletion of SatS relieves growth defects of a SatS defective mutant and affects secretion of SapM;

2) Mce1, Mce1E and Mce4A-HA require SatS for transport;

3) SatS is able to associate with target proteins such as SapM and cause disaggregation in manner akin to chaperones;

4) Structural analysis of the SatS C-terminus revealed a hydrophobic core, much like other known secretion chaperones, despite having a more unique overall structure;

5) Collective findings identify SatS as a chaperone in mycobacteria that is able to associate with and stabilize its targets for secretion.

Essential revisions:

1) As a general comment, ensure that appropriate loading controls are included in all western blots. Also, quantification of all blots would provide a numeric value of secretion and allow for statistics. This should be done. The *sapM* mutant analysis of cell associated and secreted proteins should be combined into one figure. The loading controls in Figure 2 are irrelevant if there isn't a positive control. Later in Figure 4 the authors show the cell-associated proteins; all of these data should be combined from a single experiment (i.e., with the same cultures), examining cell associated and secreted proteins along with loading controls.

2) "Virulence" really can't be measured in an ex vivo macrophage assay and in the absence of in vivo data, refrain from using this term. The only conclusion that can be made is that bacterial growth is affected. Also, are all of the strains plasmid-transformed i.e. the wild type and mutant strains should have empty vectors to rule out plasmid effects for all of the reported assays, but in particular for the macrophage work? These controls must be included.

3) The authors neglect the SecA1 system in their interpretations throughout the manuscript. It appears that SatS could also be a chaperone for the SecA1 system and the substrates of SecA2, that the authors focused on, could also be substrates of SecA1. As the *satS* null strain is more defective than the *secA2* null strain in SapM and Mce secretion (Figure 2), does this mean that SatS functions as a chaperone for both SecA1 and SecA2 systems and SapM and Mce proteins are secreted by both? In the third paragraph of the subsection “SatS as a protein export chaperone”, the authors state that the lower protein levels of the substrates in the *satS* mutant compared to the *secA2* mutant explains why the *satS* mutant secretion defect was more dramatic than that of the *secA2* mutant. But if the substrates depend on the SecA2 system for secretion, shouldn't they not be secreted at all in the null? More careful exposition of these findings and appending conclusions would be useful.

4) The models proposed in the Discussion are not supported by the data presented. The authors argue against a role for SatS in general SecA2 functioning because they don't observe an effect of the *satS* null mutant on all SecA2 substrates. Instead, the authors favour a model where in order for SecA2 to be delivered or engage the SecYEG channel, it must first be bound to a substrate and SatS functions as a protein export chaperone that facilitates this SecA2-substrate interaction. If this is true, why can't the SatS-independent SecA2 substrates cause accumulation of the SecA2 K129R mutant in the membrane in the absence of SatS? Another possibility is that as a chaperone, SatS increases the protein stability of the SecA2 K129R mutant and the *satS* null results in an unstable SecA2 K129R protein that does not interfere with SecA1 system. The authors should show the western blots used to make the calculations in Figure 1D, which may address this possibility. This potential function of SatS may or may not be relevant when WT SecA2 is expressed. It is possible that the K129R mutation is causing the protein to be unfolded and targeting it to SatS.

5) Why doesn't complemented SatS mutant look like H37Rv in Figure 2A?

---

## [Author Response]

Essential revisions:1) As a general comment, ensure that appropriate loading controls are included in all western blots. Also, quantification of all blots would provide a numeric value of secretion and allow for statistics. This should be done. The sapM mutant analysis of cell associated and secreted proteins should be combined into one figure. The loading controls in Figure 2 are irrelevant if there isn't a positive control. Later in Figure 4 the authors show the cell-associated proteins; all of these data should be combined from a single experiment (i.e., with the same cultures), examining cell associated and secreted proteins along with loading controls.

We addressed these comments with the following actions:

– We quantified all of our immunoblots for monitoring secretion as requested (Figures 2, 3, 6 and 9) from triplicate experiments. Quantitated mean values and statistical significance compared to the wild type strain are now provided in these figures (mean values and asterisk indicating significance are provided under a representative blot).

– To address the missing loading control, which was a loading control for secreted proteins of *M. smegmatis*, we added an immunoblot for the secreted Mpt32 protein to Figures 2, 6, and 9. To immunoblots for ruling out cell lysis contamination of secreted culture filtrate proteins (CFP) we added positive controls (Figure 2—figure supplement 3).

– We remade CFP and whole cell lysates from the same cultures for Figure 2. These new data are presented Figure 2A and 2E for *M. tuberculosis* and Figure 2C and 2G for *M. smegmatis*. Importantly, the results are identical to what we presented previously.

– We rearranged the presentation of the Results to allow for secreted and cell associated protein analysis to be discussed together. This change required reorganizing text and Figures 2, 3, and 4.

2) "Virulence" really can't be measured in an ex vivo macrophage assay and in the absence of in vivo data, refrain from using this term. The only conclusion that can be made is that bacterial growth is affected. Also, are all of the strains plasmid-transformed i.e. the wild type and mutant strains should have empty vectors to rule out plasmid effects for all of the reported assays, but in particular for the macrophage work? These controls must be included.

We addressed these comments with the following actions:

– We replaced the term virulence and now more precisely describe the results as demonstrating that SatS is required for growth in macrophages.

– In all experiments, wild type and mutant strains carry empty vectors. A sentence was added to the Materials and methods clarifying this point. For the macrophage experiments, we modified the figure to indicate the presence of empty vectors in wild type and mutant strains.

3) The authors neglect the SecA1 system in their interpretations throughout the manuscript. It appears that SatS could also be a chaperone for the SecA1 system and the substrates of SecA2, that the authors focused on, could also be substrates of SecA1. As the satS null strain is more defective than the secA2 null strain in SapM and Mce secretion (Figure 2), does this mean that SatS functions as a chaperone for both SecA1 and SecA2 systems and SapM and Mce proteins are secreted by both? In the third paragraph of the subsection “SatS as a protein export chaperone”, the authors state that the lower protein levels of the substrates in the satS mutant compared to the secA2 mutant explains why the satS mutant secretion defect was more dramatic than that of the secA2 mutant. But if the substrates depend on the SecA2 system for secretion, shouldn't they not be secreted at all in the null? More careful exposition of these findings and appending conclusions would be useful.

We apologize for not clearly addressing the partial nature of *secA2* mutant export defects and for not discussing before the possibility of SatS also working with SecA1 and the general Sec pathway. A consistent finding in all published studies of *secA2* mutants across three mycobacterial species is that the export defects of *secA2* mutants is never 100% (i.e. in the absence of SecA2 some residual export occurs) (Braunstein et al., 2001; van der Woude et al., 2014; Feltcher et al., 2015; Zulauf et al., 2018). Although the pathway responsible for the residual export in a *secA2* mutant is unknown, SecA1/the general Sec pathway is the likely candidate. Thus, SatS may also work with SecA1. Furthermore, with our studies so far, we cannot rule out the possibility of there being substrates of SatS that are completely SecA2-independent. We modified the Discussion to cover these details.

4) The models proposed in the Discussion are not supported by the data presented. The authors argue against a role for SatS in general SecA2 functioning because they don't observe an effect of the satS null mutant on all SecA2 substrates. Instead, the authors favour a model where in order for SecA2 to be delivered or engage the SecYEG channel, it must first be bound to a substrate and SatS functions as a protein export chaperone that facilitates this SecA2-substrate interaction. If this is true, why can't the SatS-independent SecA2 substrates cause accumulation of the SecA2 K129R mutant in the membrane in the absence of SatS? Another possibility is that as a chaperone, SatS increases the protein stability of the SecA2 K129R mutant and the satS null results in an unstable SecA2 K129R protein that does not interfere with SecA1 system. The authors should show the western blots used to make the calculations in Figure 1D, which may address this possibility. This potential function of SatS may or may not be relevant when WT SecA2 is expressed. It is possible that the K129R mutation is causing the protein to be unfolded and targeting it to SatS.

We addressed the comments with the following actions:

– The reviewers raise a good point about our model for suppression by loss of *satS*. This question about our model is now addressed in the Discussion. We suspect the reason deleting *satS* is sufficient to suppress *secA2 K129R* phenotypes is that the threshold for phenotypic suppression does not require all SecA2 K129R to be diverted from the SecYEG channel.

– In response to the possibility raised of SatS increasing the protein stability of SecA2 K129R, we added the requested representative immunoblot to Figure 1D. This immunoblot demonstrated that SecA2 K129R levels are not affected by the presence or absence of SatS, which argues against this proposal for SatS acting as a chaperone that stabilizes SecA2 K129R. We also added text to the Results section indicating that the level of SecA2 K129R level was unchanged in a *satS* mutant.

5) Why doesn't complemented SatS mutant look like H37Rv in Figure 2A?

Our SatS complementing plasmid used in *M. tuberculosis* only restored SatS levels to 26% of wild type levels, which likely accounts for the partial complementation observed. We added this SatS immunoblot data in Figure 2—figure supplement 3A and we now note the partial complementation in the Results section.